# How measurement noise limits the accuracy of brain-behaviour predictions

Martin Gell [1,2] ✉, Simon B. Eickhoff [2,3], Amir Omidvarnia [2,3], Vincent Küppers [2], Kaustubh R. Patil [2,3], Theodore D. Satterthwaite [4], Veronika I. Müller [2,3,5] & Robert Langner [2,3,5] ✉

Major efforts in human neuroimaging strive to understand individual differences and find biomarkers for clinical applications by predicting behavioural phenotypes from brain imaging data. To identify generalisable and replicable brain-behaviour prediction models, sufficient measurement reliability is essential. However, the selection of prediction targets is predominantly guided by scientific interest or data availability rather than psychometric considerations. Here, we demonstrate the impact of low reliability in behavioural phenotypes on out-of-sample prediction performance. Using simulated and empirical data from four large-scale datasets, we find that reliability levels common across many phenotypes can markedly limit the ability to link brain and behaviour. Next, using 5000 participants from the UK Biobank, we show that only highly reliable data can fully benefit from increasing sample sizes from hundreds to thousands of participants. Our findings highlight the importance of measurement reliability for identifying meaningful brain–behaviour associations from individual differences and underscore the need for greater emphasis on psychometrics in future research.

Major ongoing efforts in human neuroimaging research aim to understand individual differences and identify biomarkers for clinical applications. One particularly promising approach in this regard is the prediction of clinically relevant phenotypes in individuals (e.g. symptoms, treatment response, intellectual abilities) from functional and structural brain measurements[1–3]. Patterns of resting-state functional connectivity, defined as the statistical relationship (typically Pearson's correlation) between regional time courses of brain activity, have been widely used as brain features for the prediction of behavioural phenotypes[4,5]. Much of previous research has focused on developing and improving such predictive modelling approaches[6–8]. However, thus far, accuracies have remained too low to provide major insights into neural substrates of individual differences in behaviour or reach clinical relevance[9–13].

An essential prerequisite for identifying replicable brain–behaviour associations is sufficient reliability of measurements[14,15]. In psychometrics, reliability broadly reflects the consistency of scores across replications of a testing procedure (Standards for Educational and Psychological Testing, 2014)[16]. In the context of individual differences research, test–retest reliability has received the most attention, as it reflects the degree to which a measure ranks individuals consistently across multiple occasions (i.e. low performers remain low performers on repeated testing). Note that this assumes the measure in question assesses a stable characteristic of the individual or the amount of change between occasions does not differ between individuals (e.g., due to practice from repeated testing). Test–retest reliability is typically evaluated by intraclass correlation (ICC), which is the ratio of between-subject

[1]Department of Psychiatry, Psychotherapy and Psychosomatics, Medical Faculty, RWTH Aachen University, Aachen, Germany. [2]Institute of Neuroscience and Medicine (INM-7: Brain & Behaviour), Research Centre Jülich, Jülich, Germany. [3]Institute of Systems Neuroscience, Medical Faculty and University Hospital Düsseldorf, Heinrich Heine University Düsseldorf, Düsseldorf, Germany. [4]Department of Psychiatry, Perelman School of Medicine, Penn Lifespan Informatics and Neuroimaging Center, University of Pennsylvania, Philadelphia, PA, USA. [5]These authors contributed equally: Veronika I. Müller, Robert Langner. ✉e-mail: m.gell@fz-juelich.de; r.langner@fz-juelich.de

variance and total variance, composed of between-subject, within-subject and error variances (see McGraw and Wong[17] for a detailed discussion). Measurement noise, understood as the random variability that produces a discrepancy between observed and true values (or repeated observations), is therefore tightly related to reliability as it contributes to error variance in the calculation of ICC. That is, a high level of noise results in low reliability if the between-subject variance is held constant. ICC can range from 1 to 0 and is often interpreted as excellent for ICC > 0.8, good for 0.6–0.8, moderate for 0.4–0.6 and poor for <0.4[18,19].

While a large amount of focus has been put on assessing the reliability of brain-based measures[20–22] and ways to improve them[7,23–27], the reliability of behavioural assessments used as prediction targets has been largely neglected. Selecting scientifically or clinically relevant targets for prediction is often guided by pragmatism and logistic constraints (e.g., dataset availability), rather than considerations of reliability or criterion validity. Furthermore, classical experimental paradigms collected in many studies may not be well suited for investigating individual differences as between-subject variance in such paradigms is often low by design, resulting in low reliability[18]. Finally, current assessments of the test–retest reliability of behavioural measures commonly used in the literature show that most fall below the 'excellent' reliability[18,28] that is required for clinical applications[19,29–31]. A recent meta-analysis by Enkavi and colleagues (2019) showed the median reliability of 36 tasks assessing self-regulation was on the border of good and moderate (ICC = 0.61), and newly collected data for the same tasks showed even poor reliability (ICC = 0.31). Similarly, assessments of reliability in large datasets and longitudinal samples have reported lower estimates than those reported in test manuals, which often report reliability assessed over relatively short retest intervals[32–34].

Low measurement reliability is problematic as it attenuates existing relationships between variables. In statistical analyses (e.g. correlation), this is manifested by lowering the upper bound on maximum identifiable effect size[35]. In the context of machine learning, low reliability can have a profound impact on model performance by lowering signal-to-noise ratio. Label or target noise (akin to measurement noise) reduces the accuracy of classification algorithms[36] and increases uncertainty in parameter estimates, training time[37] as well as the complexity of a given problem[38]. As a consequence, if reliability is too low, models may fit variance of no interest (e.g., measurement noise) during training. This, in turn, results in poor generalisation performance or a failure to learn altogether. Therefore, low out-of-sample prediction accuracy may be a consequence of unreliable targets rather than a weak underlying association. This, in turn, can hamper the investigation of brain–behaviour relationships and strongly undermine efforts directed at biomarker discovery.

Due to effect size attenuation, low reliability also increases the sample sizes necessary to identify effects[39,40]. Similarly, targets with higher measurement noise require larger training sets to achieve comparable classification accuracy to less noisy targets[41,42]. As a consequence, the estimated strength of brain associations with behavioural phenotypes will be attenuated and require very large samples to become stable[43]. These considerations make large datasets for biomarker discovery a necessity rather than an advantage, which in turn poses undesirable logistical, financial and ethical challenges.

Here, we investigate how test–retest reliability of behavioural phenotypes impacts their predictability in typical studies of brain–behaviour relationships. Using a simulation approach and empirical data from four large-scale datasets, we show that low reliability systematically reduces out-of-sample accuracy when predicted

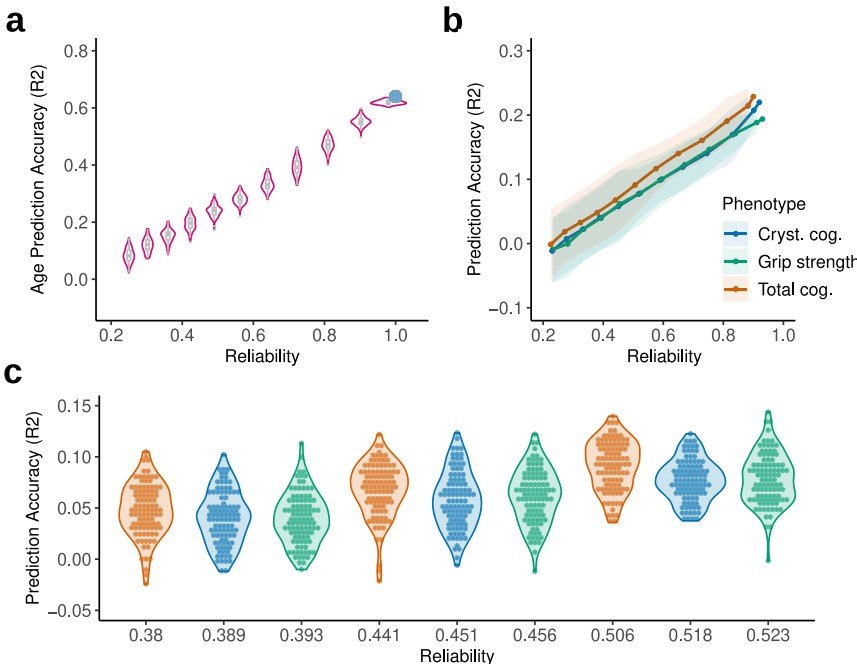

**Fig. 1 | Impact of reliability on prediction accuracy in the HCP-A dataset.**
**a** Impact of directly reducing the reliability of age on prediction accuracy (amount of target score variance explained by predicted scores as indicated by $R^2$). Each boxplot summarises the accuracy of predicting 100 simulated datasets within each reliability band and is centred at the median, with the bounds representing the interquartile range and whiskers the min/max values. **b** Impact of reducing the correlation between original and simulated target scores (reflecting reduced reliability) on accuracy in prediction of total cognition composite score, crystallised cognition composite score and grip strength. *X*-axis was adjusted for individual behaviour reliability. Solid lines represent the mean accuracy across all 100 simulated datasets in each reliability band, shaded areas represent 2 standard deviations in accuracies. **c** Effect of random noise on variability in prediction accuracy. The colour legend is common for panels **b** and **c**.

from functional connectivity. Furthermore, using a sample of 5000 adults, we illustrate the tradeoff between reliability and sample size.

## Results

### Low phenotypic reliability reduces the accuracy of brain–behaviour predictions

To systematically test the impact of target reliability on out-of-sample prediction accuracy, we simulated behavioural assessments with reduced test–retest reliability using empirical data from the Human Connectome Project Aging dataset (HCP-A) as a basis. Reliability was manipulated by incrementally increasing the proportion of random noise within the target variable.

As a proof of principle, we first present results for participant age prediction ($n = 647$). As expected, systematically reducing the reliability of age resulted in a sharp decrease in accuracy as measurement noise increased (Fig. 1a). Crucially, every 0.2 drop in reliability reduced the coefficient of determination ($R^2$) on average by 25%. Mean absolute error (MAE) and correlation of predicted and observed scores followed a similar pattern (Supplementary Results Fig. 1). The observed rate of change in accuracy replicated in the UK Biobank dataset (UKB; Supplementary Results Fig. 2) and was robust to variations in parcellation (Supplementary Results Fig. 3) and algorithm choice (Supplementary Results Fig. 4). Reducing the reliability of resting-state functional connectivity by shortening scan duration reduced the overall prediction accuracy, but did not impact the pattern of change in $R^2$ (Supplementary Results Figs. 5 and 6).

Next, we investigated the attenuation of prediction accuracy that can be expected in typical studies of brain–behaviour associations by systematically adding noise to the most reliable measures (ICC ≈ 0.9) available in the HCP-A dataset ($n = 550$, Supplementary Table 4). This way we simulated new phenotypes with reliabilities that are common in neuropsychological assessments and have plausible true effect sizes. Total cognition could be predicted with an accuracy of $R^2 = 0.23$ (MAE = 10.37), crystallised cognition with $R^2 = 0.22$ (MAE = 10.24) and grip strength with $R^2 = 0.19$ (MAE = 9.79). Similarly to age, reducing their reliability resulted in a decrease in prediction accuracy (Fig. 1b). For all three assessments, $R^2$ halved when simulated data reached reliability of approximately 0.6 ($R^2_{total\ cog.} = 0.12$; $R^2_{crystalized\ cog.} = 0.1$; $R^2_{grip\ strength} = 0.1$). Importantly, analysis choices such as confound regression, feature space or feature reliability resulted in small variations in prediction accuracy on empirical and simulated data but had no impact on the rate at which performance decreased (Supplementary Results Figs. 7–9). For MAE and correlation between predicted and observed scores, see Supplementary Results Fig. 10.

We note that prediction accuracy could vary by 0.1 $R^2$ or more between the best and worst-performing simulated datasets for the same level of noise. When reliability reached 0.5 or lower, a value not uncommon for behavioural assessments, such variability could lead to large fluctuations in accuracy that would warrant different conclusions regarding the success of predictions (Fig. 1c). For example, in grip strength prediction, the accuracy at reliability $r = 0.46$ ranged from $R^2 = -0.01$ to $R^2 = 0.12$ acorss all hundred models. All results were corrected by the reliability of phenotypes estimated in previous work ($ICC_{total\ cog.} = 0.9$[44]; $ICC_{crystalized\ cog.} = 0.86$[44]; $ICC_{grip\ strength} = 0.93$[45]). As these can vary between studies, we also provide uncorrected results assuming perfect reliability of phenotypes to display more general trends in Supplementary Results Fig. 11.

### Target reliability is related to prediction accuracy

Next, we directly investigated the relationship between reliability and brain–behaviour prediction accuracy in empirical data where reliability could be estimated. Using test–retest data from the Human Connectome Project dataset Young Adult ($n = 46$; HCP-YA) and follow-up data from the UKB dataset ($n = 1890$), we estimated the reliability of 36 behavioural assessments in HCP-YA (ICCs =

0.25–0.89; median ICC = 0.63; Supplementary Results Fig. 12) and 17 assessments in UKB (ICCs = 0.22–0.81; median ICC = 0.54; Supplementary Results Fig. 13). The resulting reliability was then correlated with their prediction accuracy in the respective full sample (HCP-YA = 771; UKB = 5000) (Fig. 2).

Based on our simulation results, we expected an increasing attenuation of prediction accuracy as assessment reliability decreased. Confirming this, $R^2$ displayed a substantial correlation with test–retest reliability in the HCP-YA dataset ($r(34) = 0.62$, $p < 0.001$, 95% CI [0.37, 0.79]) and the UKB dataset ($r(15) = 0.65$, $p < 001$, 95% CI [0.25, 0.86]) even though retest intervals were longer (mean retest = 2 years and 6 months compared to 5 months in HCP-YA). This was also replicated in the ABCD dataset despite likely developmental effects on reliability and prediction accuracy ($r(23) = 0.85$, $p < 0.001$, 95% CI [0.69, 0.93]; Supplementary Results Fig. 14). Given the small number of retest participants ($n = 46$) in HCP-YA, we also correlated $R^2$ with the lower and upper bounds of the ICC and observed the same relationship ($r(34) = 0.54$, $p < 0.001$, 95% CI [0.25, 0.74] and $r(34) = 0.61$, $p < 0.001$, 95% CI [0.36, 0.78], respectively). As models with negative $R^2$ values may not be comparable in accuracy, we also correlated only models

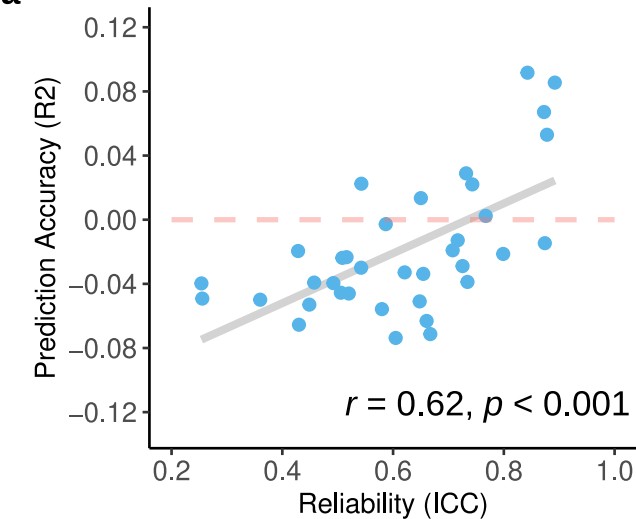

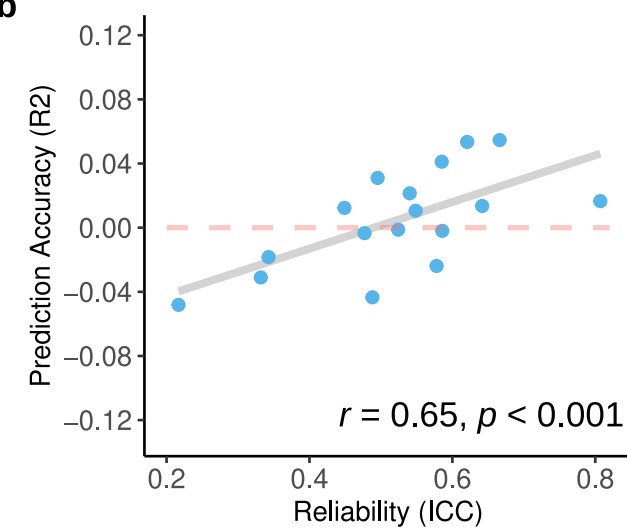

**Fig. 2 | Association between reliability and prediction accuracy. a** HCP-YA and (**b**) UKB dataset. Each data point represents a behavioural assessment in each dataset. Correlations between reliability and prediction accuracy are indicated in the plot. For degrees of freedom and confidence intervals, please see the text.

## Age

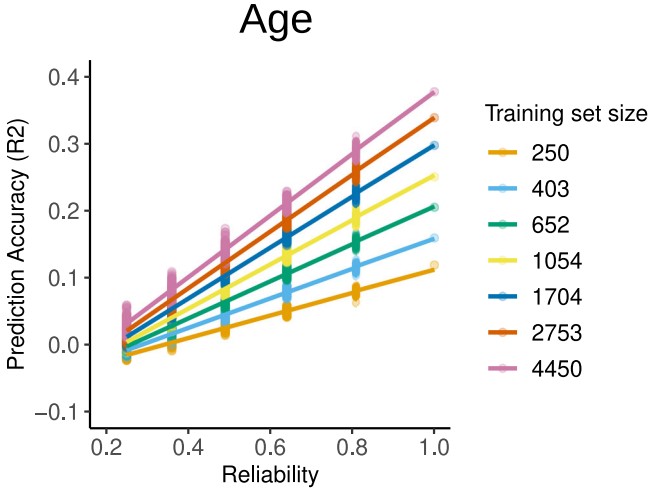

## Grip strength

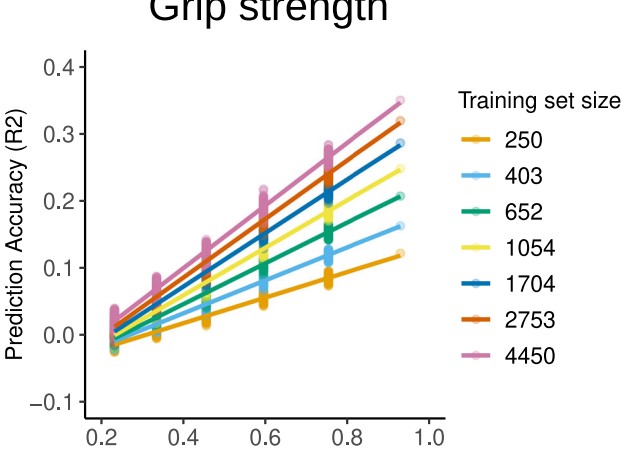

**Fig. 3 | Prediction and subsampling in UKB.** Impact of training set size on original and simulated data with reduced reliability. Results were fitted with a linear function for illustration purposes.

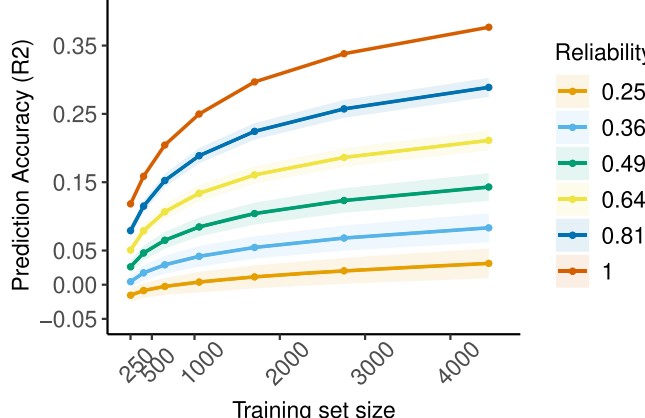

**Fig. 4 | Improvement in prediction accuracy scales with sample size in simulated data.** Impact of training set size on age prediction accuracy in empirical and simulated data with varying levels of reliability. Solid lines represent the mean accuracy across all 100 simulated datasets in each reliability band and shaded areas represent 2 standard deviations in accuracies.

with positive $R^2$ with reliability in HCP-YA and found an even stronger correlation ($r(7) = 0.71$, $p = 0.032$, 95% CI [0.09, 0.93]). Similar to our main analysis, all variables with reliability lower than <0.6 displayed very low accuracy ($R^2 < 0.02$). Conversely, only variables with excellent reliability (the picture vocabulary task, total cognition, grip strength, reading English and crystallised cognition) could achieve $R^2 > 0.05$.

### Influence of phenotype reliability on prediction accuracy scales with sample size

Finally, we sought to investigate how the interaction between reliability and sample size impacts brain–behaviour prediction. Using 5000 participants from the UKB dataset, we repeated the same simulation approach in geometrically spaced training set sizes ranging from $n = 250$ to 4450. We were only able to systematically increase random noise in two example phenotypes—age and grip strength—as none of the cognitive assessments exhibited reliability high enough to warrant manipulating it (we illustrate one such example in Supplementary Results Fig. 15).

Systematically increasing noise resulted in reduced accuracy for all training set sizes and followed the same pattern of $R^2$ halving for every 0.4 drop in reliability observed in our previous analysis (Fig. 3). Importantly, a change of 0.2 in reliability had a larger impact on prediction performance than a change in training set size (e.g., from $n = 1054$ to $n = 1704$). For age prediction, even samples of 652

participants with excellent reliability ($r = 0.81$) produced comparable accuracy to the full sample ($R^2_{mean} = 0.15$, $R^2_{sd} = 0.005$) with moderate reliability ($r = 0.49$) that is common across behavioural assessments ($R^2_{mean} = 0.14$, $R^2_{sd} = 0.01$). This effect was less pronounced for phenotypes displaying weaker association with functional connectivity, where more reliable training sets required at least half the size of the less reliable sample to achieve comparable accuracy (Supplementary Results Fig. 15).

Increasing sample size always resulted in higher prediction accuracy irrespective of reliability. However, the largest improvements in accuracy were observed for highly reliable data, while data with moderate reliability showed only minor gains (Fig. 4). This was particularly pronounced for samples below 1000 participants. Next, we investigated how prediction accuracy of empirical data with varying levels of reliability increases as a function of training set size (Fig. 5). Replicating results from simulated data with reduced reliability in Fig. 4, phenotypes with excellent reliability ($ICC_{grip\ strength} = 0.81$; $ICC_{age} \approx 1.0$) displayed a steeper and larger improvement in accuracy as sample size increased. Phenotypes with good reliability ($ICC_{Associative\ learning} = 0.62$; $ICC_{Cognitive\ flexibility} = 0.67$; $ICC_{Fluid\ intelligence} = 0.64$) showed only minor changes in accuracy with proportionally smaller improvements (Supplementary Results Table 1). These remained unchanged when the maximum training set size was increased by an additional 2500 participants (Supplementary Results Fig. 16).

## Discussion

Here we demonstrate the burden of low behavioural test-retest reliability on out-of-sample prediction performance in brain–behaviour associations. Our results suggest that, especially when associations between brain features and behavioural assessments are weak to moderate, levels of reliability that are common for behavioural phenotypes can substantially attenuate large portions of shared variance. Importantly, this attenuation holds irrespective of feature definition, prediction algorithm or dataset, suggesting that analytical choices have little impact. Furthermore, we show that while a larger sample size increases the accuracy of brain–behaviour predictions, highly reliable data in smaller samples can produce comparable results to large amounts of moderately reliable data and depending on the size of the true relationship, can even outperform it. Following on from these findings, we show that only highly reliable data can fully benefit from increasing sample sizes from hundreds to thousands of participants.

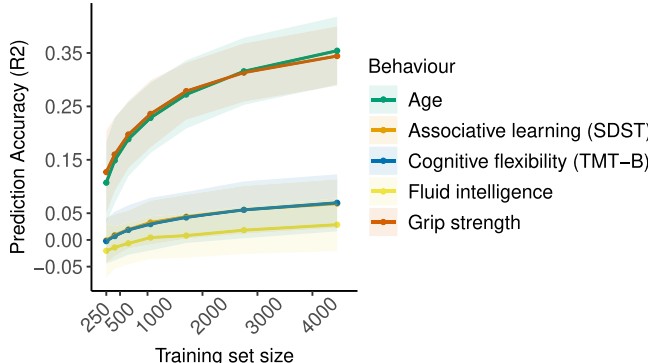

**Fig. 5 | Improvement in prediction accuracy scales with sample size in empirical data.** Impact of training set size on prediction accuracy of empirical behaviours. Solid lines represent the mean accuracy across 100 subsamples and shaded areas represent 2 standard deviations in accuracy. SDST symbol digit substitution test, TMT-B trail-making task part B.

## Phenotypic reliability is important for robust results

The attenuation of a correlation between two variables by their reliability was already described by Charles Spearman in 1910. Here we aimed to demonstrate that machine learning approaches widely used to identify brain–behaviour associations also suffer from low phenotypic reliability and show its impact on out-of-sample prediction accuracy. Generally, we found that reliability attenuated out-of-sample prediction accuracy similarly to what has been described for in-sample correlation[15,39,40] and classification[36,46]. Building on arguments emphasising the importance of reliability in biomarker research[14], we illustrate the amount of attenuation that can be expected by the reliability of routinely collected neuropsychological assessments available in most large-scale neuroimaging datasets. Our results suggest that moderate reliability (ICC = 0.6–0.4) can produce serious attenuations of prediction accuracy irrespective of the dataset, rs-fMRI reliability and analytical choices. Specifically, for ICC = 0.6, we observed prediction accuracy on average half of that when the same variable had ICC ≈ 0.9. Moreover, even good levels of reliability (ICC = 0.6–0.8) were found to substantially attenuate brain–behaviour associations. Strong relationships (e.g. age) were equally susceptible to strong attenuation but, unlike weaker ones, could still be predicted with poor reliability. However, current estimates indicate that such large effect sizes for brain–behaviour associations are the exception rather than the rule[43,47]. Overall, these results indicate that high test-retest reliability of behavioural phenotypes is crucial to fairly evaluate the potential of neuroimaging in predicting individual differences in behaviour.

Supporting previous literature[18,28,32,34,48,49], most behavioural assessments in the datasets used here (HCP-YA, UKB and ABCD) showed reliabilities within the good to the moderate range that were found to be susceptible to large attenuations (median ICC = 0.51; Supplementary Results Fig. 17), despite desirable levels for clinical applications[29,31]. As many large neuroimaging datasets utilise similar measurement instruments (e.g. NIH Toolbox[50]), low prediction accuracies observed in many recent reports may be partly driven by suboptimal reliability of prediction targets[7,25,51–55]. This, in turn, limits further insights into interindividual differences in brain function and the search for neuroimaging-based biomarkers. Importantly, our results also suggest that the field can benefit substantially from improving measurement practices and optimising behavioural reliability to increase SNR for predictive modelling and increasing association effect sizes.

The final attenuation of brain–behaviour relationships will be determined by the joint reliability of both neuroimaging features and behavioural targets[39,56]. The reliability of functional connectivity

depends on the network[57], preprocessing steps[22] and scan duration, with longer acquisition leading to greater reliability[22,58,59]. The marked difference in age prediction accuracy between HCP and UKB datasets we observed here, is, therefore, likely related to differences in rsfMRI acquisition (6 min in UKB compared to 26 min in HCP-A), in addition to lower precision in reported age in the UKB (measured in years compared to months in HCP). In other words, low phenotypic reliability that produced serious attenuation in the HCP-A dataset is likely to display even greater attenuation in datasets with less reliable fMRI measurements. Therefore, the results shown here may represent an optimistic scenario for the field, as 26 min of resting-state images collected over two days is, especially in clinical settings, uncommon. However, we also emphasise that the impact of low phenotypic reliability generalised across datasets as well as when feature reliability was directly manipulated. Therefore, even with exceptionally reliable fMRI measurements, unreliable phenotypes are still likely to substantially attenuate out-of-sample prediction accuracy, as consistent ranking across individuals is impaired.

In addition to overall low prediction performance for data with less than good reliabilities, we observed a large variance in prediction accuracy in simulated data. Specifically, datasets with moderate and poor reliability showed accuracies that could result in opposite conclusions. For example, at ICC = 0.45, the highest accuracies ($R^2 \approx 0.1$) were comparable to those reported for many behavioural assessments[60], while the worst observed accuracy represented a failure of prediction (i.e., $R^2 < 0$). As in our simulations, measurement noise was randomly distributed; these results suggest that even phenotypes with moderate reliability may contain enough noise to produce results that will not replicate. Conversely, the higher the reliability, the lower the risk of the variance in results caused by random noise to reach $R^2 = 0$. Our findings, therefore, reinforce the necessity for authors to follow best-practice guidelines, replicate their predictions and validate their models in truly independent samples or datasets[61].

## Large samples are necessary but not sufficient

In a recent study, Marek and colleagues (2022) have suggested that investigating brain–phenotype associations requires sample sizes of $n > 2000$, as sampling variability in small effects can result in imprecise effect size estimates. While cognitive ability and total psychopathology used by the authors as exemplary phenotypes have been reported to have excellent reliability (ICC > 0.9; however, see Tiego & Fornito[62] for a discussion), the remaining phenotypes that were assessed have more modest reliabilities (ICC = 0.31–0.82)[32–34,48,63]. Given this large variation in reliability, the reported sample size requirement is likely not a one-size-fits-all recommendation[64], as increasing the reliability of many collected behavioural measurements will result in larger effect sizes, effectively reducing the sample size requirement.

Here we demonstrate that depending on the true association strength, highly reliable phenotypes can reach comparable prediction accuracy using samples in the hundreds rather than thousands, as they are less subject to marked attenuation by low reliability. These results suggest that collecting more reliable data may be particularly important for research questions (assuming cross-sectional design is appropriate) where many thousands of participants are difficult to acquire (e.g., specific conditions) and discuss ways to implement this below. However, more importantly, we demonstrate that only reliable phenotypes can fully benefit from observed improvements in prediction accuracy as training set sizes increase from hundreds to thousands of participants[65–67]. Conversely, measurements with poor reliability are likely suboptimal candidates for big data initiatives, as collecting thousands of participants will only yield minor increases in accuracy before saturating. Therefore, improving the measurement reliability of appropriately selected phenotypes for associations with neuroimaging features will likely boost predictive (and statistical) power in large datasets. Finally, we note that our findings should not be

taken to justify the use of small *n* studies under the guise of high measurement quality. As long as true associations between behavioural phenotypes and neuroimaging display small effect sizes, very large samples will be necessary to estimate them. Thus, it is important that on top of considering measurement reliability, researchers continue to follow guidelines for generalisable[68] and reproducible predictive modelling[61,69–71].

Across a broad range of tested variables, empirical reliability (estimated from the datasets) was rarely excellent (5 out of 36 tested in HCP-YA, 0 out of 17 in UKB and 0 out of 25 in ABCD), replicating previous observations[32]. Furthermore, empirical reliability was generally lower than that reported after test development[44,45,50,72]. Similar differences in reliability between different datasets are not uncommon[32–34,49] and may be due to differences in retest intervals. However, assessments of behaviour in large datasets, in particular, may be subject to other sources of measurement noise resulting from specifics of big data collection, such as site differences, staff training, relatively low number of trials designed to lower the burden on participants or shortened versions of validated assessments, and participant fatigue from lengthy acquisition protocols. At the same time, best practices in assessing test–retest reliability during test development are not always adhered to, likely producing further discrepancies between studies[73]. We further note that the test-retest reliability of many measures in large datasets is currently hard to assess, as outside of the HCP-YA none of the other datasets assessed here (HCP-A, UKB and ABCD) or many other large openly available datasets have dedicated test-retest samples. The inability to assess phenotype reliability in these datasets precludes the possibility of disentangling whether a poor model performance in a given study is due to measurement error or truly reflects a low effect size. If phenotype reliability is indeed substantially lower in large datasets than that reported at test development, then many available datasets may be of limited use for individual-differences research; and additionally, further, increasing sample sizes (e.g. to biobank levels) without considering psychometrics will be of little benefit. We, therefore, urge that, moving forward, any attempts at identifying biomarkers must involve careful consideration and thorough assessment of the reliability of behavioural as well as neuroimaging measurements (e.g., in re-test samples) before data is collected at larger scales and evaluated for predictive power.

## Improving phenotypic reliability

A wealth of previous literature has discussed ways of improving measurement reliability. Prior to the acquisition, this can be achieved by opting for a deeper phenotyping design, either in the laboratory by introducing more rigorous testing strategies and collecting more trials per participant (for an overview see ref. 74) or by means of ecological momentary assessment[75], taking measures to increase between-subject variance[76], or acquiring multiple assessments for data aggregation[56]. In already acquired data, researchers should select relevant measurements with the best psychometric properties. For behavioural phenotypes, assuming that error variance and loading of all items on a latent dimension are equal[77], data reduction techniques such as principal component analysis or summary scores can increase reliability and lead to larger effect sizes than individual items[13,43,60,78,79]. Supporting this, composite scores of the NIH toolbox tasks in the HCP datasets were more reliable than individual assessments and reached higher prediction accuracy. Similarly, averaging left and right-hand grip strength in the UKB dataset compared to each hand separately leads to improvement in both reliability and accuracy. Comparable increases can also be achieved when grip strength is averaged across testing occasions (Supplementary Results Fig. 18). If equal item loading on a latent dimension cannot be assumed, reliability can be increased using latent modelling frameworks that account for systematic and unsystematic errors. However, more work is necessary to identify the most cost-effective strategies for optimising the reliability of both brain and behavioural measurements without sacrificing measurement validity. To this end, future research should focus on ways to improve the reliability of already acquired data and evaluate best practices to preserve reliability when acquiring new data at large scales.

Although the high reliability of either measurement is necessary for meaningful investigations of prediction accuracy, it is not sufficient. For instance, highly reliable phenotypes that do not capture a valid representation within the brain are not likely to improve effect sizes. Moreover, many behavioural measurements are validated against other established psychological scales or with specific populations in mind, rather than developed in light of their biological relevance. As a result, they may not be well-suited for investigations of brain behaviour associations, and thus, enhancing their reliability may bring little improvement in effect size. Similarly, structural MRI metrics that display better reliability than functional connectivity[80,81], are often poorer predictors of many psychological constructs[79] that may instead rely on intrinsic fluctuations in neural activity[82]. Therefore, while optimising measurement reliability offers one possible avenue for improving the investigation of individual differences, it will not guarantee larger effect sizes[83] or better prediction accuracy, especially if the selection of appropriate phenotypes is neglected.

In conclusion, the recent availability of large-scale neuroimaging datasets, combined with advances in machine learning, has enabled the investigation of population-level brain–behaviour associations. In this study, we demonstrate that common levels of reliability across many behavioural phenotypes in such datasets can strongly attenuate or even conceal actual associations. This, in turn, can lead to scientifically questionable conclusions about the predictive potential of neuroimaging and hinders clinical translation. Therefore, greater emphasis needs to be placed on refining behavioural phenotyping in large datasets on top of similar efforts directed at neuroimaging. Together, more reliable neurobiological measurements and markers of behaviour will be necessary to fully exploit the benefits of big data initiatives in neuroscience, promote the identification of potential biomarkers, and contribute to reproducible science.

**Table 1 | Overview of datasets and samples used in main analyses**

| Dataset | Analysis | Sample (Female)[a] | Age in years | Age at follow-up |
|---|---|---|---|---|
| HCP-A | Prediction of simulated data and selected phenotypes | 647 (351) | 60.2 (±0.14) | |
| HCP-YA | Prediction of all phenotypes | 771 (358) | 28.5 (±3.7) | |
| | Test–retest | 46 (32) | 30.2 (±3.4) | 30.6 (±3.3) |
| UKB | Prediction of simulated data and all phenotypes | 5000 (2714) | 63.6 (±7.3) | |
| | Test–retest | 1890 (1012) | 61.1 (±7) | 63.5 (±6.9) |
| ABCD | Prediction of all phenotypes | 4133 (2123) | 10 (±0.6) | |
| | Test–retest | 2102 (1026) | 10 (±0.63) | 11.9 (±0.64) |

*HCP-A* Human Connectome Project Aging, *HCP-YA* Human Connectome Project dataset Young Adult, *UKB* UK Biobank.
[a]Participant sex was self-reported.

## Methods

### Ethical approval

The reanalysis of openly available data was approved by the ethics committee of the Medical Faculty at Heinrich Heine University Düsseldorf (4039 and 2018-317-RetroDEuA). Each dataset used in this study had obtained ethical approval by their respective ethics committees. Participants in all datasets gave informed written consent and were compensated by the respective studies and collection sites.

### Datasets

We utilised data from four large-scale datasets (Table 1). Noise simulations were done using data from the Human Connectome Project Aging dataset (HCP-A) due to its favourable ratio between imaging data quality (see Supplementary Methods Table 2 for dataset comparison) and variance in phenotypic data with high reliability. The Human Connectome Project dataset Young Adult (HCP-YA), UK Biobank (UKB) and Adolescent Brain Cognitive Development (ABCD) were used to investigate the association between reliability and prediction accuracy as test-retest or follow-up behavioural data was available in all datasets (and not in HCP-A). Finally, the UKB sample was used to investigate the interaction between reliability and sample size, given the large number of participants available.

**Human Connectome Project Aging dataset.** For our primary simulation analysis, we used data from the Human Connectome Project Aging dataset[84,85], obtained from unrelated healthy adults. Only participants with all four complete runs of resting-state fMRI (rs-fMRI) scans and no excessive head movement (framewise displacement <0.25 mm, which corresponded to 3 SD above the mean) were analysed, resulting in a sample of 647 participants for age prediction (351 female, ages = 36–89) and ~550 who had all phenotypic data of interest available (see Supplementary Table 3 for exact n for each phenotype).

The HCP scanning protocol involved high-resolution T1w MRI images that were acquired on a 3 T Siemens Prisma with a 32-channel head coil using a 3D multi-echo MPRAGE sequence (TR = 2500 ms, 0.8 mm isotropic voxels). The rs-fMRI images were acquired using a 2D multiband gradient-echo echo-planar imaging (TR = 800 ms, 2 mm isotropic voxels). Four rs-fMRI sessions with 488 volumes each (6 min and 41 s) were acquired on two consecutive days, with one anterior-to-posterior and one posterior-to-anterior encoding direction acquired on each day.

**Human Connectome Young Adult dataset.** To investigate the relationship between reliability and prediction accuracy we used data from the Human Connectome Project Young Adult dataset[86], partly consisting of related healthy participants. Only participants with all four complete runs of rs-fMRI, no excessive head movement (framewise displacement <0.3 mm, which corresponded to a displacement of 3 SD above the mean) and all phenotypes of interest were included (n = 713, 358 female, ages = 22–35). In total, 36 behavioural phenotypes that were available for all participants and did not display strong ceiling effects were selected for prediction (see Supplementary Table 4 and Fig. 19 for a full list of phenotypes and their distributions). Standardised scores were used when available. Additionally, a test–retest dataset for participants with all 36 assessments (n = 46, 32 female, ages = 22-35) was used to estimate phenotypic reliability.

The HCP scanning protocol involved high-resolution T1w MRI images that were acquired on a 32-channel head coil on a 3 T Siemens "Connectome Skyra" scanner using a 3D single-echo MPRAGE sequence (TR = 2400 ms, 0.7 mm isotropic voxels). The resting state fMRI images were acquired using whole-brain multiband gradient-echo echo-planar imaging (TR = 720 ms, 2 mm isotropic voxels). Four rs-fMRI sessions with 1200 volumes each (14 min and 24 s) were acquired on two consecutive days, with one left-to-right and one right-to-left phase encoding direction acquired on each day.

**UK Biobank.** To investigate the association between prediction accuracy and reliability as well as how reliability interacts with sample size, we randomly sampled 5000 (2714 female, ages = 48–82) participants from all healthy participants in the UK Biobank sample[87]. Healthy participants were defined as participants without lifetime prevalence of cerebrovascular diseases, infectious diseases affecting the nervous system, neuropsychiatric disorders or neurological diseases based on ICD-10 diagnosis from hospital inpatient records and self-report (see Supplementary Methods Table 5 for all excluded data fields). All participants had complete rs-fMRI scans and displayed no excessive head movement (framewise displacement <0.28 mm, which corresponded to a displacement of 3 SD above the mean). Within this sample, we selected 17 phenotypes that were available for all participants and did not display strong ceiling effects (see Supplementary Table 6 and Fig. 20 for a full list of phenotypes and their distributions). Of those, age and grip strength were used for creating simulated data. Additionally, a sample of 1890 (1012 female, ages = 48–79) participants with available follow-up data for all 17 phenotypes from the follow-up imaging session was used to estimate phenotypic reliability. The mean interval between the initial imaging session and the follow-up session was 2 years and 6 months.

The UKB scanning protocol[88] included structural and resting state fMRI images acquired at four imaging centres (Bristol, Cheadle Manchester, Newcastle and Reading) with harmonised Siemens 3 T Skyra MRI scanners with a 32-channel head coil. T1w MRI images were acquired using a 3D MPRAGE sequence (TR = 2000 ms, 1.0 mm isotropic). One rs-fMRI session with 490 volumes each (6 min and 10 s) was acquired using a multiband echo-planar imaging (TR = 735 ms, 2.4 mm isotropic voxels).

**Adolescent brain cognitive development.** To investigate if our association between phenotype reliability and prediction accuracy generalises to an additional dataset with different preprocessing we used data from the Adolescent Brain Cognitive Development study[89] baseline sample from the ABCD BIDS Community Collection[90]. Only English-speaking participants without severe sensory, intellectual, medical or neurological issues and all available behavioural phenotypes were used (see Supplementary Table 7 for a full list). Furthermore, all participants had to have complete rs-fMRI data and pass the ABCD quality control for their T1 and resting-state fMRI. This resulted in a total of 4133 participants (2123 female, ages = 9–11). Additionally, a sample of 2102 (1026 female, ages = 9–11) participants with available follow-up data for all phenotypes from the first follow-up session was used to estimate phenotypic reliability. The mean interval between the initial imaging session and the follow-up session was 1 year and 11 months.

The ABCD acquisition protocol[91] was harmonised across 21 sites on Siemens Prisma, Phillips, and GE 750 3 T scanners. It included high-resolution T1w MRI images with a 32-channel head coil using a 3D multi-echo MPRAGE sequence (TR = 2500 ms, 1.0 mm isotropic voxels). The rs-fMRI images were acquired using gradient-echo echo-planar imaging (TR = 800 ms, 2.4 mm isotropic voxels) and included two sessions totalling 20 min.

### Simulation of different levels of reliability of selected phenotypes

As increasing noise for the purposes of our analyses may only be meaningful in highly reliable phenotypes, we selected prediction targets in the HCP-A dataset based on their published estimates of reliability: age (ICC ≈ 1.0), grip strength (ICC = 0.93; Reuben et al.[45]), total cognition composite (ICC = 0.86–0.95; Akshoomoff et al.[72]; Heaton et al.[44]) and crystallised cognition composite (ICC = 0.9; Akshoomoff et al.[72]; Heaton et al.[44]). In the UKB dataset, we only manipulated noise in age (ICC ≈ 1.0) and grip strength (ICC = 0.93–0.96; Bohanon et al. (2011); Hamilton et al. (1994)), as none of the cognitive assessments

exhibited reliability values that were high enough to warrant lowering it with noise (the highest reliability we found was for the trail-making B task[48] at $r = 0.78$). For each of the selected prediction targets, we created simulated datasets with varying amounts of noise. According to classical measurement theory[92], any measurement reflects a mixture of the measured entity and random (as well as systematic) measurement noise. The reliability of a variable can thus be reduced by increasing the proportion of error or noise variance while holding between-subject variance constant, thereby reducing the signal-to-noise ratio. Here we manipulated only the unsystematic measurement noise, defined as random variability that produces a discrepancy between observed and true values (or repeated observations). Increasing random noise is ideal for investigating test–retest reliability as it only affects the variability of measurements around the mean and thus manipulates the ranking across individuals.

In order to induce increasing levels of noise in the target variable, we created datasets that correlated with the originally observed (empirical) targets at a pre-specified Pearson's correlation. This method was chosen to increase the interpretability of the resulting attenuation of brain–behaviour associations by controlling the amount of noise. The data generation procedure was as follows: First, a random vector was sampled from a standard normal distribution with the same mean and standard deviation as the original empirically acquired data (in the HCP these were age-adjusted and normalised to mean = 100 and SD = 15). Next, we calculated the residuals of a least squares regression of the sampled vector ($X$) on the empirical data ($Y$). The resulting orthogonal vector representing the portion of $X$ that is independent of $Y$ was then again combined with the original empirical data $Y$ through scaling by the pre-specified correlation. This adjustment process manipulated the relative contributions of $Y$ and the residuals of $X$ on $Y$ in the resulting simulated vector. The formula used for this process was:

$$X_{Y\rho} = \rho\sigma(Y\perp)Y + \sqrt{1 - \rho^2}\sigma(Y)Y\perp \tag{1}$$

where $X_{Y\rho}$ is the new 'simulated' vector that correlates with the empirical data $Y$ at a predefined correlation $\rho$. $Y\perp$ represents the residuals of a least squares regression of a randomly sampled vector $X$ against $Y$. All simulations were created using custom code in R [version 4.0.4] and are provided online[93].

The pre-specified correlations for simulated data based on the HCP-A dataset were set to correlate with the original data at $r = 0.99$, 0.95, 0.9, 0.85, 0.8, 0.75, 0.7, 0.65, 0.6, 0.55 and 0.5. Given the high computational load for large samples, simulated UKB data were set to correlate at $r = 0.9$, 0.8, 0.7, 0.6 and 0.5 with the original data. For each level of correlation, simulation was repeated 100 times, thus totalling 4400 simulated datasets for HCP-A (4 assessed phenotypes × 11 noise levels × 100 repeats) and 1500 simulated datasets for UKB (3 assessed phenotypes × 5 noise levels × 100 repeats). Simulated datasets were scaled and offset to have approximately the same mean and standard deviation as the original measurements to facilitate absolute agreement (i.e. stability across repeated measurements) between the original data and the simulated test–retest data in order to harmonise test–retest correlations and ICC. As age did not follow a normal distribution, we first estimated its probability density from the original data and then sampled simulated data from this distribution instead.

## Phenotype preprocessing

As we used linear ridge regression for prediction, all phenotypes that displayed a right-skewed distribution were transformed with a natural log transform. As this procedure manipulated data within participants, there was no data leakage across participants.

## fMRI preprocessing

Both HCP datasets provided minimally preprocessed data. The preprocessing pipeline has been described in detail elsewhere[94]. Briefly,

this included gradient distortion correction, image distortion correction, registration to participants' T1w image and to MNI standard space followed by intensity normalisation of the acquired rs-fMRI images, and independent component analysis (ICA) followed by an ICA-based X-noiseifier (ICA-FIX) denoising[95,96]. Additional denoising steps were conducted by regressing mean time courses of white matter and cerebrospinal fluid and the global signal, which has been shown to reduce motion-related artefacts[97]. Next, data were linearly detrended and bandpass filtered at 0.01–0.1 Hz.

The UKB data were preprocessed through a pipeline developed and run on behalf of UK Biobank[98] and included the following steps: motion correction using MCFLIRT[99]; grand-mean intensity normalisation of the entire 4D fMRI dataset by a single multiplicative factor; highpass temporal filtering using Gaussian-weighted least-squares straight line fitting with sigma = 50 s; Echo Planar Imaging unwarping; Gradient Distortion Correction unwarping; structured artefact removal through ICA-FIX[95,96]. No low-pass temporal or spatial smoothing was applied. The preprocessed datasets (i.e. filtered_func_data_clean.nii in the UK Biobank database) were normalised to MNI space using FSL's applywarp command.

The ABCD dataset was preprocessed ABCD-BIDS pipeline as part of the ABCD-BIDS Community Collection (ABCC; Collection 3165), which has been described in detail elsewhere[90]. The pipeline included distortion correction and alignment using Advanced Normalisation Tools (ANTS), FreeSurfer segmentation, and surface as well as volume registration using FSL FLIRT rigid-body transformation. Processing was done according to the DCAN BOLD Processing (DBP) pipeline, which included de-trending and de-meaning of the rs-fMRI data, denoising using a general linear model with regressors for tissue classes and movement. The data were then bandpass filtered between 0.008 and 0.09 Hz using a second-order Butterworth filter. DPB respiratory motion filtering (18.582–25.726 breaths per minute), and censoring (frames exceeding an FD threshold of 0.2 mm or failing to pass outlier detection at ±3 standard deviations were discarded) were then applied.

## Functional connectivity

The denoised time courses from all datasets were parcellated using the Schaefer atlas[100] with 400 cortical regions of interest for all main analyses. The signal time courses were averaged across all voxels of each parcel. Parcel-wise time series were used for calculating functional connectivity between all parcels using Pearson correlation. For HCP datasets, the correlation coefficients of individual sessions (4 per participant) were transformed into Fisher-$Z$ scores, and for each connection, an average across sessions was calculated. To investigate the robustness of our results to granularity and parcellation selection, functional connectivity between denoised time courses of 200, 300 cortical regions from the Schaefer atlas[100] as well as 300 cortical, subcortical and cerebellar regions of interest defined by Seitzman et al.[101] was calculated. Regions were modelled as 6-mm spheres and calculated from resting state data from the HCP Aging dataset. Finally, to investigate the generalisation of our results to another dataset with different preprocessing steps, the ABCD dataset was parcellated using HCP's 360 ROI atlas template[102].

## Prediction

We used the scikit-learn library [version 0.24.2[103]] to predict all target variables from functional connectivity using custom code available online[93]. Accuracy was measured using coefficient of determination ($R^2$), mean absolute error (MAE) and Pearson correlation between predicted and observed target values. The $R^2$ represents the proportion of variance (in the target variable) that has been explained by the independent variables in the model and was calculated as:

$$R^2(y,\hat{y}) = 1 - \frac{\sum_{i=1}^{n}(yi - \hat{y}i)^2}{\sum_{i=1}^{n}(yi - \bar{y}i)^2} \tag{2}$$

where $\hat{y}_i$ is the predicted value of the ith sample and $y_i$ is the corresponding true value for total n samples. $\bar{Y}$ Represents the mean across all $y$. In this formulation, the $R^2$ is not interchangeable with the correlation coefficient squared. All predictions were performed using linear ridge regression as it showed a favourable ratio of computation time to accuracy in previous work[104] and preliminary testing (see Supplementary Fig. 21). Out-of-sample prediction accuracy was evaluated using a nested cross-validation with 10 outer folds and 5 repeats. Hyperparameter optimisation (inner training folds) of the $\alpha$ regularisation parameter for ridge regression was done using efficient leave-one-out cross-validation[105]. The model with the best $\alpha$ parameter was then fitted on the training folds and tested on the outer test folds. Within each training fold, neuroimaging features were standardised by z-scoring across participants before models were trained in order to ensure that individual features with large variance would not dominate the objective function. Before prediction (of both original and simulated data), participants with target values 3 SD from the sample mean were removed from the complete sample to minimise the impact of extreme values resulting from random sampling in simulated data. As a preprocessing step prior to training, neuroimaging features were z-scored within participants.

**Control analyses for simulation results in HCP-A.** To verify our analyses were robust to analytical degrees of freedom, we repeated our analyses of the HCP-A dataset using support vector regression, an alternative node definition for functional connectivity features (using ROIs from ref. 101) and feature-wise confound removal. For algorithm comparison, we trained a support vector regression with a linear kernel on neuroimaging features. Out-of-sample prediction accuracy was evaluated using a non-nested cross-validation with 10 outer folds and 5 repeats. A heuristic was used to efficiently calculate the hyperparameter $C$[106]:

$$c = \frac{1}{\frac{1}{n}\sum_{i=1}^{n}\sqrt{G[i,i]}} \qquad (3)$$

where $G$ is the matrix multiplication of features and transposition of features (here: functional connectivity).

To investigate whether confounding effects impacted our results, standard confound variables (age and sex for the prediction of all phenotypes) were removed from the connectivity features using linear regression. Confound removal was performed within each training fold and the confound models were subsequently applied to test data to prevent data leakage[107].

Finally, we investigated the impact or feature reliability (here functional connectivity) on our results. As the length of the resting state time-course has been shown to influence the reliability of function connectivity[22,58,59], we reduced the amount of resting state data used for calculating it and repeated all predictions. In the main analyses of the HCP-A, all 4 resting sessions from both days were used (22 min and 44 s). To reduce feature reliability, functional connectivity was then calculated using both sessions from each day separately (13 min and 22 s) and lastly, to mirror the UKB acquisition protocol, only the very first session acquired in the anterior-to-posterior direction on day one (6 min and 41 s) was used for calculating functional connectivity. All control analyses are presented in the supplemental material.

**Association between reliability and prediction accuracy.** The relationship between target reliability and prediction accuracy (measured as $R^2$) was investigated using the HCP-YA dataset. First, the test-retest data of 46 participants was used to estimate measurement reliability for 36 different behavioural phenotypes by calculating ICC between the scores from the first and second visits. ICC was calculated using a two-way random effects model for absolute agreement, often referred to as ICC [2,1][108]. Next, all selected measures were predicted in a sample of 713 participants from the HCP-YA dataset using linear ridge regression. As the HCP-YA dataset includes related participants, cross-validation was done using a 5 times repeated leave 30% of families out approach, instead of the 10-fold random split used in other analyses. Family members were always kept within the same fold in order to maintain independence between the folds. Confounding effects of age and sex on features were removed using linear regression trained on the training set and applied to test data within the cross-validation. Finally, the resulting prediction accuracies ($R^2$) of the 36 different phenotypes were correlated with their corresponding reliability (calculated from the test–retest data) and tested for significance (using a two-tailed test). To validate our findings, the above-described approach (with the exception of cross-validation) was repeated using the UKB dataset. Reliability was estimated for 17 different behavioural assessments using ICC2 between measurements collected during the first and follow-up imaging visits in 1893 participants. All phenotypes were predicted in a set of 5000 participants from the UKB using ridge regression in nested cross-validation with 10 outer folds and 5 repeats used for our main analyses. Correlations between reliability and prediction accuracy were not corrected for multiple comparisons.

**Subsampling procedure and prediction in the UKB dataset.** To examine how the effects of reliability on prediction performance interact with increasing sample size, we randomly sampled geometrically spaced samples (series with a constant ratio between successive elements) from 5000 participants of the UK Biobank starting from $n = 250$ (250, 403, 652, 1054, 1704, 2753, 4450). By doing so, we aimed to cover sample sizes ranging from those available in larger neuroimaging studies to international consortia levels. To be able to compare prediction accuracy between different sample sizes we used a learning curve function from Sklearn. In this approach, we first partitioned a test set of 10% of the full sample (i.e. $n = 500$). From the remaining data, geometrically spaced samples (250, 403, 652, 1054, 1704, 2753, 4450) were sampled without replacement. Each subsample was then used to train a ridge regression model with hyperparameter optimisation using the same cross-validation set-up with 10 outer folds and 5 repeats used in previous analyses. This approach made the comparison of accuracy between different sample sizes possible as the test set is held constant for all training samples. The entire procedure was repeated 100 times for all simulated and empirical data.

**Reporting summary**
Further information on research design is available in the Nature Portfolio Reporting Summary linked to this article.

## Data availability
This study utilised publicly available data from the UK Biobank (https://www.ukbiobank.ac.uk/enable-your-research), the HCP Young Adult (https://www.humanconnectome.org/study/hcp-young-adult), HCP Aging (https://www.humanconnectome.org/study/hcp-lifespan-aging/data-releases), and ABCD (https://nda.nih.gov/study.html?id=2313). ABCD and HCP Aging study data are available under restricted access (https://nda.nih.gov/abcd/request-access) to researchers with an approved NDA Data Use Certification (DUC). Similarly, to access data from the UK Biobank, researchers are required to comply with a data use agreement and apply for the data resource (https://www.ukbiobank.ac.uk/register-apply/).

## Code availability
All scripts and computational resources utilised in this manuscript, including exemplary data, can be accessed in a public repository[93] and found online at: https://doi.org/10.5281/zenodo.13901196.

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

## Acknowledgements

This research has been conducted using the UK Biobank Resource under Application Number 41655. Funding was provided by the Deutsche Forschungsgemeinschaft (DFG, German Research Foundation)—269953372/GRK2150; 69953372/GRK2150, EI816/11-1; and 431549029/SFB1451, Jülich-Aachen Research Alliance (JARA), the National Institute of Mental Health (R01-MH074457), the Helmholtz Portfolio Theme "Supercomputing and Modelling for the Human Brain", and the European Union's Horizon 2020 Research and Innovation Programme under Grant Agreement No. 945539 (HBP SGA3).

## Author contributions

Martin Gell: Conceptualisation, investigation, formal analysis, writing—original draft, writing—review and editing; Simon B. Eickhoff: Conceptualisation, resources, supervision, project administration, funding acquisition, writing—review and editing; Amir Omidvarnia: Resources, software, data curation, writing—review and editing; Vincent Küppers: Validation, software, data curation, writing—review and editing; Kaustubh Patil: Methodology, software, supervision, writing—review and editing; Theodore D. Satterthwaite: Resources, supervision, funding acquisition, writing—review and editing; Veronika I. Müller: Conceptualisation, methodology, writing—review and editing, supervision, project administration; Robert Langner: Methodology, project administration, writing—review and editing.

## Funding

## Competing interests

The authors declare no competing interests.
