## [Peer Review File · Nature Communications]

Reviewers' Comments:

Reviewer #1:

Remarks to the Author:

The overarching goal of this paper is to demonstrate how reliability and sample size affect predictive accuracy for brain-behavior associations. The authors focus on the effect of reliability on prediction of behavioral or other phenotypic measures using fMRI data. Generally, the authors tackle an important problem that researchers using fMRI to study individual differences have recently been grappling with, and the results provide useful guidance to researchers in this area. However, there are a few points of confusion and clarifications that would be helpful to address. In addition, the conclusions seem overly broad given the results, as discussed below.

1. “Overall, our findings suggest that research programmes focused on identifying brain-behaviour associations from individual differences must seriously consider the reliability of outcome measures.” While this is true, it seems to suggest that this hasn’t already been considered in the literature. But there is a long history of discussion of this topic, going back to the 2009 Vul et al. paper. In addition, there is nothing special about outcome measures; reliability needs to be considered for all measures (brain, behavior, or other) that are used in the context of individual differences (regardless of the specific analytic approach); see more on this below.

2. The following statement is misleading: “reliability reflects how accurately a test can measure a specific construct”. As noted in-text, reliability is the consistent ranking of an individual as it relates to a score. As defined by Slaney (2017, pg. 33), reliability is “... broadly defined as a quantitative index of the degree of measurement precision associated with a test (or subtest) score”. Or, alternatively, Standards for Educational and Psychological Testing define reliability as “... the consistency of scores across replications of a testing procedure” (2014, pg. 33). The concept of *validity* is generally taken to describe how well a particular measure indexes a particular construct. Importantly, one can have a measure that is highly reliable (i.e. consistent across measurement replications) but still fails to validly measure a particular construct. More generally, discussion of measurement concepts in the manuscript needs to be much more crisp.

3. The use of the concept of “predictive validity” in the manuscript is somewhat problematic, because it implies validity with respect to an underlying construct, rather than the goodness of fit of a statistical model to data. For example, let’s say that a researcher is able to strongly predict age from resting functional connectivity, but that this turns out to be due to motion confounds that are not properly addressed. To say that the model has “predictive validity” is to imply that there is a conceptual

relationship between the brain measure and age, when in reality the true construct being measured is motion rather than brain function. As I understand it, predictive validity in the measurement literature refers to whether a measurement, e.g., score from depression measure, can predict some other conceptually related criterion. Eg., Standards of Psychology Testing defines predictive validity as “how accurately test data collected at one time can predict criterion scores that are obtained at a later time.”. This is also consistent with the definition in from others (Cronbach & Meehl, 1955, DOI: 10.1037/h004095). Perhaps “predictive accuracy” or “predictive power” is a more appropriate label for the concept being addressed in the present manuscript?

4. The authors use two different samples that involve different acquisition protocols and preprocessing decisions. Can the authors report this information in a supplementary table so the reader can easily follow how each data was acquired and which preprocessing decisions were used? A table with these details would also help delineate how many subjects had what types of data for which phenotypes/samples. Currently, it is hard to track.

5. There are differences in both analysis and acquisition between the samples, which could in theory result in differences in the reliability of the resulting imaging phenotypes. It would be useful for the authors to comment on the potential impact of these differences.

6. The crux of the argument in the manuscript concerns the reliability of the *phenotypes*. But the attenuation will affect both measures, in this case the outcome measure (e.g., age or cognition) and functional connectivity. Several publications have discussed the issues regarding reliability in rsfMRI (Noble et al., 2019, DOI: 10.1016/j.neuroimage.2019.116157) and multivariate prediction in Spisak et al (2022, DOI: 10.1038/s41586-023-05745-x) and Tervo-clemmens et al. (2023, DOI: 10.1038/s41586-023-05746-w). If the prediction accuracy halves when going from 1 to .80, why would we even expect accuracy as high as .64 for age when the second measure, functional connectivity, likely has reliability between .2 - .5? How should we make sense of the interpretations given the interaction between phenotypic and brain reliabilities (i.e., Hedge et al discuss both measures' reliabilities)? Moreover, if the issue here is that the predictive accuracy decreases due to reliability, if the authors have a highly reliable measures (e.g., age = 1.0, grip strength = .93), wouldn't their point be strengthened if they demonstrate how these items have stable prediction *across* rsfMRI measurement occasions?

7, The authors note that “highly reliable data in smaller samples can outperform large amounts of moderately reliable data”. I worry that this claim will be leveraged in service of underpowered correlation/prediction analyses. Most interesting psychological

phenotypes are going to realistically have correlations with brain phenotypes that are relatively low (cf. Marek et al., 2022). Let's say that we are studying a brain-behavior relationship where the true underlying correlation in the population is $r=0.2$ (which is well above the maximum correlations observed by Marek). A sample size of 193 is required for 80% power to find this association in a single test, assuming perfect reliability; of course, a researcher will usually want to look across multiple phenotypes, which will reduce power substantially due to multiple test corrections and thus require much larger samples. While I agree that higher reliability can improve power, I implore the authors to not provide researchers with an excuse to do badly powered studies.

8. In the discussions, the authors state: "Our results show that high reliability of phenotypes is paramount for the prediction of individual differences from neuroimaging" & "Excellent reliability of phenotypic assessments is paramount for investigating brain-behaviour associations.". This may be a bit overstated for a couple of reasons. First, the evidence shows how model performance may be attenuated from the perspective of the outcome variable. Little investigation/discussion of rsfMRI measurement is provided. Furthermore, no repeated prediction across rsfMRI occasions is included. In light of low reliability in rsfMRI (see Noble et al., 2019), from "we observed that reliability levels lower than 0.6 can make the investigation of predictive validity (e.g. is functional connectivity a good predictor of phenotype X) meaningless" do the authors really think that that most, if not all, predictive utility analyses using rsfMRI will be "meaningless". Second, to achieve "perfect" reliability may not always make sense. This relates to a point the authors noted in the introduction "this assumes the measure in question assesses a stable characteristic of the individual or the amount of change between occasions does not differ between individuals (e.g., due to practice from repeated testing)." Perfect reliability scores of function/structure are likely critical in certain circumstances, such as the use of fMRI for neurosurgical planning (as noted in Bennett & Miller, 2010, DOI: 10.1111/j.1749-6632.2010.05446.x). However, this may not always be reflective of the process/system being measured, which may fluctuate in the real-world. For example, in developmental work reliabilities that are .90 or higher, in some cases, may be problematic. Why collect 3 waves of scores when they are extremely stable? In fact, Cronbach's alphas that are .94+ are presumed to contain redundancies—often highlighting that the measure is oversaturated with items. So the reliability of measure may depend on the context and research question(s). One wouldn't expect low within-subject variance on IQ scores and lack of shifts in between-subject variances as a function of environmental/developmental factors. This may instead suggest that the type of questions being asked and how the data are being modeled should be modified. Alternatively, without fluctuations on certain measures, characterizing changes in people across time would be a futile effort. Rather than writing off the entirety of a research program (e.g., all behavioral phenotypes are insufficient for analysis of rsfMRI

associations), it would be helpful to contextualize the problem in units of measurement of different phenotypes and fMRI. Through this, perhaps the authors can offer some recommendations that researchers can use to navigate two highly unreliable measures?

9. Regarding the statement “composite or summary scores are likely to lead to better prediction accuracy than individual assessments”, this is not always true. Sum scores of multidimensional measures may alter/misrepresent effects and may not be an appropriate way to capture certain phenotypes (McNeish & Wolf, 2020, DOI: 10.3758/s13428-020-01398-0), because sum scores/averages impose a measurement model that suggests each loading and error is equal. In general, latent modeling frameworks, in the presence of multi-scale or multi-session development, can capture the latent construct in a manner that accounts for systematic/unsystematic error. How the latent value is derived depends on whether the assumption is that the construct exists in nature (e.g. reflective) or whether the latent construct depends on the scholar’s construction/interpretation of the construct (e.g., formative; Coltman et al., 2008, DOI: 10.1016/j.jbusres.2008.01.013). That latent process can help with improved accuracy in representing the latent variable and improvement in its reliability, but if misspecified, may also misrepresent the scores like the sum score. Again here it would be helpful to express the measurement issues in a more sophisticated way.

Reviewer #2:

Remarks to the Author:

This is a timely investigation on an important topic of phenotypic measurement reliability in brain-based multivariate prediction of psychological and health outcomes. Strengths of the study are clear: generally well-considered methodological choices, multiple independent datasets, and a cogent (and very welcomed) integration of psychometric theory and neuroimaging research. Strengths here, however, are not without some concerns and areas for further clarification. Most notably, clarifying that the methods but critically the results and implications here reflect a situation of making often low reliability phenotypes even less reliable (based on the simulation approach). Additionally, it would benefit the manuscript to include increased integration of psychometric research outside of neuroimaging (ie, scale development) and a bit information on the data generation model for simulation.

1. In general, the authors appropriately state that the version of “reliability” here is not synonymous with reliability in the standard sense. However, some of these demarcations become very important with respect to the implications of the study and future recommendations. In general, the analyses reported here in focus on a “proof of

principle” demonstrations that added measurement noise decreases prediction accuracy, but the actionable information moving forward is less clear to me. In the simplest terms these results simulate adding noise and what happens, ultimately however, we as a field want to reduce noise and increase measurement reliability. That is that the “best” highest reliability results presented here are actually the current state of brain-phenotype relations, progressively adding noise makes things worse but it’s not always clear under which circumstances this matters.

In more detail, consider the case of $r(\text{original}, \text{simulated}) = .8$, if the true phenotypic reliability is .5, it’s a bit unclear to me how to interpret the magnitude of what $r(\text{orig}, \text{sim})$ of .8 really means here in absolute terms with regard to actionable information on the phenotypic reliability. Can the authors clarify this? At minimum I agree with the authors rationale, that it would seem that simulated vector with $r(\text{orig}, \text{sim}) .8$ for a given measure with an existing phenotypic reliability of .5 would index reliability of even lower than .5 of course (as an extra noise process is being added). Yet, while the authors mention this “Thus, the ‘true’ reliability of simulated phenotypes is lower than the preset level of correlation between simulated and original data, as this preset level will be attenuated by each variable's original reliability.”, it is unclear how these magnitudes influence absolute judgements and inferences and general recommendations based on the reported results. For example, on the age variable (which the authors rightfully point out is likely most consistent with reliability as age is considered measured with nearly zero error), one can draw a general recommendation that reliability matters (added noise decreases predictive accuracy). But the influences on other phenotypes (e.g. Fig 1 Cryst Cog, Grip Strength, Total Cog) with presumably a range of general “to-be-predicted effect sizes” is less clear given this magnitude interpretation challenge. For example $r(\text{orig}, \text{simulated}) = 1$ in these plots represents the measures with the current (less than ideal reliabilities they have), progressively adding noise here makes them worse yes, but the authors also mention that we should be cautious of our current measures. I think a full simulation-based approach that disaggregates the “measures at hand” and whatever range of ~multivariate predictive effect sizes therein from the issues of reliability would provide a much stronger base to make recommendations. That being said, I appreciate the complexity of this given the high dimensionality of the imaging data and the predictive algorithms. In general, further clarifying the issue of “adding noise” to an already questionably reliable measure, the range of possible ~multivariate predictive effect sizes, and limitations on what recommendations can be concretely made moving forward (recognizing that many of these simulations represent making low reliability measures even worse) is necessary.

2. The authors do well to be clear on the goal “relative reliability” for phenotypic selection in the current work, but I would encourage a further consideration of general

psychometric work. For example, in many ways at the stage of phenotypic selection in large scale existing datasets “the die has been cast” regarding phenotypic measurement selection and by extension the associated psychometric properties therein. That is, that, as the authors partially allude to but I would consider ways to further increase this, individual phenotypic measures via scale development are often designed or “validated” with psychometric targets in mind. There are of course incentives at play in scale development and I appreciate the authors alluding to distinctions between reliabilities reported in scale development and those observed in real large-scale data, but I would encourage the authors to consider how the individual measures themselves are developed and what we should do with “end of the pipeline use cases” (relative selection of phenotypes). I can imagine more concrete recommendations being that 1. We need to improve measurement reliability generally (although its relative impact on minimal sample sizes isn’t entirely clear see point 1 above) and 2. When looking at large-scale data we should emphasize selecting the most reliable phenotypic measures (the inferences here are based on the relative selection of phenotypes). Both general ideas appear in the manuscript but deserve further emphasis particularly in the discussion section.

3. Can the authors provide a bit more information on the data generation model for the simulated data? I can imagine solutions for such a data generation model that creates vectors for a given correlation with the original, but it would be useful to include for example relevant formulae/proofs or more detail on the procedure.

Reviewer #3:

Remarks to the Author:

In the manuscript, "The Burden of Reliability: How Measurement Noise Limits Brain-Behaviour Predictions", Gell and colleagues demonstrate how imperfect reliability of behavioral phenotypes can result in reduced associations between the brain and behavior. In-and-of-itself this is not a novel concept (attenuation of correlation [Spearman, 1904]), but the authors did a nice job of demonstrating how this phenomenon plays out in simulation and real data from the HCP and UK Biobank datasets. I think their main argument that "only highly reliable data can fully benefit from increasing sample sizes from hundreds to thousands of participants" was well supported. Thus, the paper was methodologically well done. However, my excitement around this topic was dampened for several reasons. For one, improved measurement reliability was highlighted in the Marek + TC BWAS paper as an avenue needed for a more reproducible science, so I am not sure of the novelty of this paper beyond that. There are also several other papers that have come out over the last year showing the

same (e.g., Nikolaidis et al., 2022; Tiego & Fornito, 2022). The specific argument that high reliability is a prerequisite for robust brain-behavior relationships is likewise not novel (Vul et al., 2009; Milham et al., 2021).

1. The authors mention in the Discussion that "Large samples are necessary but not sufficient" for improved brain-behavior reproducibility. However, within that section, the authors appear to contradict themselves by saying "Here we demonstrate that reliable phenotypes may be predicted using samples in the hundreds rather than thousands, as highly reliable measurements are not subject to attenuation by reliability". The latter clause is a truism, and in the first clause the authors mistakenly imply that a measure with high reliability (e.g., shoe size) will ensure a large enough effect size with the brain to detect it with a small sample size. This may be true for age, but it is certainly not true for all non-brain phenotypes.

2. Although I agree 100% that improved reliability of behavioral measures is needed, the elephant in the room - even in this paper - is clearly sample size. One could argue from these results that both reliability and thousands of individuals are needed for improved reproducibility in BWAS studies. For example, in Fig. 3a (prediction of age), a training sample size of ~400 results in a prediction accuracy of ~0.10, whereas a sample of ~4,400 boosts it to ~0.40. As the authors show, reliability and prediction accuracy are tightly linked, but so is sample size and prediction accuracy. To me, this is overwhelming evidence that we need both large sample and reliability. This needs to come through more clearly in the text.

3. Difficult to acquire data. The authors note, "Particularly for research questions where thousands of participants are difficult to acquire (e.g. specific conditions), optimising for greater measurement reliability may be of greater benefit than increasing sample size. Here again, the authors are "either or" in their thinking. What if the measure is reliable, but not a valid construct in the brain? Perhaps if data are difficult to acquire for a given population, a different research design is needed other than cross-sectional brain-behavior correlations?

4. As the authors note, achieving excellent reliability (>0.9) in large consortia studies has proven difficult (5/56 across HCP and UKB in the current paper). This is not even touching on the brain data, especially resting state fMRI which certainly has poor reliability within the UKB (6 mins of data). What do the authors think this implies for existing consortia given the joint reliability of the existing brain-behavior data must be suboptimal (i.e., less than excellent)?

Reviewer #4:

Remarks to the Author:

In this manuscript, the authors show how the reliability of measured phenotypes (i.e., demographics, behavior) impacts the predictive accuracy of machine learning models trained on functional connectivity data. They show that, as reliability diminishes, so too does predictive accuracy. Importantly, they also show that, past certain levels of measurement noise, it is more effective to invest efforts into improving phenotypic reliability than to simply collect more data.

This is a timely and well-written manuscript of particular relevance in an era of large-scale neuroimaging data collection. I have no qualms with the take-home message of the manuscript, which is long-established if under-appreciated (e.g., see Bollen (1989), for a lengthy treatment of reliability on R-squared in multiple regression). I do, however, have outstanding questions that I hope the authors might consider addressing in the Discussion.

Major points

1. The observed rate at which predictive accuracy diminishes with reliability is striking. For example, in the manuscript, the authors report that $r^2 = 0.64$ of variance in age can be explained by functional connectivity. When the reliability of age is artificially diminished to 0.80, the prediction accuracy is halved to $r^2 = 0.33$. It is worth putting this in the context of Spearman's attenuation formula (Spearman, 1910): $r_{\text{obs}} = r_{\text{true}} * \sqrt{r_{xx} * r_{yy}}$, where r_{xx} and r_{yy} are the reliabilities of two variables, x and y ; r_{true} is their true (latent) correlation; and r_{obs} is the observed correlation.

For argument's sake, let us assume that the functional connectivity composite has a reliability of $r_{xx} = 0.9$ and the true correlation between this composite and age is $r_{xy} = 0.9$. Plugging these numbers in, we get: $0.9 * \sqrt{0.9 * 1.0} = 0.81$ ($r^2 = 0.65$), or about what is reported above. Now, plugging in the same numbers when the reliability of age is artificially decreased, we get: $0.9 * \sqrt{0.9 * 0.8} = 0.76$ ($r^2 = 0.58$). The variance explained in this toy example is substantially higher than what was actually observed ($r^2 = 0.58$ vs. 0.33). This begs the question as to why.

One possibility is that the multiple regression case is more complex than the bivariate case (e.g., due to interactions between imperfectly measured predictors; Bollen, 1989). A related possibility is that this reflects something specific to functional connectivity data (e.g., correlated measurement error). A third possibility is that this reflects something specific to the method (e.g., machine learning models that capitalize on chance during training perform even worse during out-of-sample prediction).

It would be useful for the authors to speculate why it is the case that even phenotypes with conventionally acceptable reliability (i.e., $\rho = 0.8$) lead to such dramatic decreases in predictive accuracy. (It would be even more helpful for the authors to demonstrate this via additional simulations, perhaps using fully simulated datasets where they can control all relevant features of the data). Such discussion might spur new research into alternative approaches to handling "suboptimal" reliability, which may prove equally fruitful as the likely substantial effort required to "acceptably" reliable behavioral measures (i.e., $\rho = 0.8$) to excellent levels of reliability (i.e., $\rho \geq 0.9$).

Minor points

2. it should be noted that one of the authors main points (i.e., only highly reliable data can fully benefit from increasing sample sizes) has been demonstrated elsewhere in psychology (e.g., Jacobucci et al., 2020; McNamara et al., 2022). Though these papers need not be discussed extensively in the manuscript, they should at least be cited as this is a problem not specific to neuroimaging.

References

Bollen, K. A. (1989). The consequences of measurement error. *Structural equations with latent variables*, 151-178.

Spearman, C. (1910). Correlation calculated from faulty data. *British Journal of Psychology*, 3(3), 271–295.

Jacobucci, R., & Grimm, K. J. (2020). Machine learning and psychological research: The unexplored effect of measurement. *Perspectives on Psychological Science*, 15(3), 809-816.

McNamara, M. E., Zisser, M., Beevers, C. G., & Shumake, J. (2022). Not just “big” data: Importance of sample size, measurement error, and uninformative predictors for developing prognostic models for digital interventions. *Behaviour Research and Therapy*, 104086.

Reviewers' comments:

Reviewer #1 (Remarks to the Author):

The overarching goal of this paper is to demonstrate how reliability and sample size affect predictive accuracy for brain-behaviour associations. The authors focus on the effect of reliability on the prediction of behavioural or other phenotypic measures using fMRI data. Generally, the authors tackle an important problem that researchers using fMRI to study individual differences have recently been grappling with, and the results provide useful guidance to researchers in this area. However, there are a few points of confusion and clarifications that would be helpful to address. In addition, the conclusions seem overly broad given the results, as discussed below.

We thank the reviewer for their helpful feedback that helped us improve our manuscript considerably. We have now clarified the points of confusion, substantially revised our discussion of psychometric concepts and focused our conclusions. Additionally, we provide additional supplementary results detailing the impact of functional connectivity reliability on our results

1. "Overall, our findings suggest that research programs focused on identifying brain-behaviour associations from individual differences must seriously consider the reliability of outcome measures." While this is true, it seems to suggest that this hasn't already been considered in the literature. But there is a long history of discussion of this topic, going back to the 2009 Vul et al. paper. In addition, there is nothing special about outcome measures; reliability needs to be considered for all measures (brain, behaviour, or other) that are used in the context of individual differences (regardless of the specific analytic approach); see more on this below.

We wholeheartedly agree and have clarified our statements as:

"Overall, our findings highlight the importance of measurement reliability for identifying brain-behaviour associations from individual differences."

We completely agree that the reliability of all measurements matters (please see our response below in points 6 and 8 about the importance of reliable features). However, we decided to only focus on the reliability of the outcome measures in our work as we believe that, in contrast to the reliability of neuroimaging, outcome measures have been mostly overlooked by neuroimagers until now. Furthermore, by only focusing on behaviour, our claims and empirical evaluations are more concrete and easier to grasp than considering all possible measures at once. However, we do see that in our previous version of the

manuscript, we put too little weight on making clear that we focus on only one side of brain-behaviour relationships and now address this shortcoming by a detailed treatment of this topic in our revised introduction and discussion.

2. The following statement is misleading: “Reliability reflects how accurately a test can measure a specific construct”. As noted in-text, reliability is the consistent ranking of an individual as it relates to a score. As defined by Slaney (2017, pg. 33), reliability is “... broadly defined as a quantitative index of the degree of measurement precision associated with a test (or subtest) score”. Or, alternatively, Standards for Educational and Psychological Testing define reliability as “... the consistency of scores across replications of a testing procedure” (2014, pg. 33). The concept of *validity* is generally taken to describe how well a particular measure indexes a particular construct. Importantly, one can have a measure that is highly reliable (i.e. consistent across measurement replications) but still fails to validly measure a particular construct. More generally, the discussion of measurement concepts in the manuscript needs to be much more crisp.

*We thank the reviewer for pointing out this potential misunderstanding of our definition of reliability. We have now rephrased the definition of reliability in lines 52-54 to **“In psychometrics, reliability broadly reflects the consistency of scores across replications of a testing procedure (2014, Standards for Educational and Psychological Testing)”**. Additionally, as detailed in our responses to points below we have attempted to clarify our discussion of measurement concepts throughout the manuscript.*

3. The use of the concept of “predictive validity” in the manuscript is somewhat problematic because it implies validity with respect to an underlying construct, rather than the goodness of fit of a statistical model to data. For example, let’s say that a researcher is able to strongly predict age from resting functional connectivity, but that this turns out to be due to motion confounds that are not properly addressed. To say that the model has “predictive validity” is to imply that there is a conceptual relationship between the brain measure and age when in reality the true construct being measured is motion rather than brain function. As I understand it, predictive validity in the measurement literature refers to whether a measurement, e.g., a score from a depression measure, can predict some other conceptually related criterion. Eg., Standards of Psychology Testing defines predictive validity as “how accurately test data collected at one time can predict criterion scores that are obtained at a later time.”. This is also consistent with the definition from others (Cronbach & Meehl, 1955, DOI: 10.1037/h004095). Perhaps “predictive accuracy” or “predictive power” is a more appropriate label for the concept being addressed in the present manuscript?

We thank the reviewer for pointing out our confusing use of the term “predictive validity”. The term was intended to denote the study of the predictive capacity of neuroimaging-derived measures. We have rephrased all instances to the suggested term of the reviewer (“prediction accuracy”) in order to keep with established definitions in measurement theory.

4. The authors use two different samples that involve different acquisition protocols and preprocessing decisions. Can the authors report this information in a supplementary table so the reader can easily follow how each data was acquired and which preprocessing decisions were used? A table with these details would also help delineate how many subjects had what types of data for which phenotypes/samples. Currently, it is hard to track.

We thank the reviewer for pointing out the unclear information on datasets. We apologize that dataset acquisition and preprocessing were not clear in the previous version, we have clarified this information in the main text Methods section under ‘Datasets’ (lines XYZ) and ‘fMRI preprocessing’ (lines XYZ) and added a supplementary table 5 with acquisition and preprocessing parameters for all 3 datasets. In the analyses relating phenotype reliability and prediction accuracy, (Figure 2) all subjects (denoted by sample size in text) had data for all predicted phenotypes. For our simulation analyses, there were small differences in available phenotypes between subjects and we already detailed how many subjects had what types of data in Supplementary Table 4. We have further revised the results section to be clearer on which phenotypes and datasets were used for which analyses (lines 174-180):

“Using test-retest data from the Human Connectome Project dataset Young Adult (n = 46; HCP-YA) and follow-up data from the UKB dataset (n = 1890), we estimated the reliability of 36 behavioural assessments in HCP-YA (ICCs = 0.25 - 0.89; median ICC = 0.63; Supplementary Results Figure 12) and 17 assessments in UKB (ICCs = 0.22 - 0.81; median ICC = 0.54; Supplementary Results Figure 13). The resulting reliability was then correlated with their prediction accuracy in the full sample (HCP-YA = 771; UKB = 5000)”

Supplementary Table 5

Acquisition and preprocessing of datasets

Parameter	Dataset		
	HCP-A	HCP-YA	UKB
Scanner	Siemens Prisma 3T	Siemens 'Connectom Skyra' 3T	Siemens Skyra 3T

Resting-state sessions	4 runs across 2 days with AP and PA phase encoding on each day	4 runs across 2 days with LR and RL phase encoding on each day	a single run acquired in AP direction
acquisition time	488 frames per run (26 min total)	1200 frames per run (58 min total)	490 volumes (6 min)
TR/TE	800/37 ms	720/33 ms	735/39 ms
Acquisition sites	4	1	4
gradient distortion correction	yes	yes	yes
intensity normalisation	yes	yes	yes
motion correction	yes	yes	yes
normalisation to MNI	yes	yes	yes
artefact removal	ICA-FIX	ICA-FIX	ICA-FIX
temporal filtering	bandpass filtered at 0.01 – 0.1 Hz	bandpass filtered at 0.01 – 0.1 Hz	highpass filtering
denoising	WM+CSF+GS regression	WM+CSF+GS regression	no

Abbreviations; AP: anterior-to-posterior; PA: posterior-to-anterior; LR: left-to-right; RL right-to-left all refer to phase encoding directions

5. There are differences in both analysis and acquisition between the samples, which could in theory result in differences in the reliability of the resulting imaging phenotypes. It would be useful for the authors to comment on the potential impact of these differences.

We agree that differences in fMRI data acquisition between the samples such as scan length will impact the reliability of the functional connectivity scores. We have now included a discussion of these differences between datasets:

“The final attenuation of brain-behaviour relationships will be determined by the joint reliability of both neuroimaging features and behavioural targets (Nunnally, 1970; Nikolaidis et al., 2022). Reliability of functional connectivity depends on the network (Tozzi, Fleming, Taylor, Raterink, & Williams, 2020), preprocessing steps (Noble et al., 2019) and scan duration, with longer acquisition leading to greater reliability (Noble et al., 2017, 2019; Cho, Korchmaros, Vogelstein, Milham, & Xu, 2021). The marked difference in age prediction accuracy between HCP and UKB datasets we observed here, is therefore, likely related to differences in rsfMRI acquisition (6 minutes in UKB

compared to 26 minutes in HCP-A), in addition to lower precision in reported age in the UKB (measured in years compared to months in HCP). In other words, low phenotypic reliability that produced serious attenuation in the HCP-A dataset is likely to display even greater attenuation in datasets with less reliable fMRI measurements. Therefore, the results shown here may represent an optimistic scenario for the field, as 26 minutes of resting-state images collected over two days is, especially in clinical settings, uncommon.”

6. The crux of the argument in the manuscript concerns the reliability of the *phenotypes*. But the attenuation will affect both measures, in this case, the outcome measure (e.g., age or cognition) and functional connectivity. Several publications have discussed the issues regarding reliability in rsfMRI (Noble et al., 2019, DOI: 10.1016/j.neuroimage.2019.116157) and multivariate prediction in Spisak et al. (2022, DOI: 10.1038/s41586-023-05745-x) and Tervo-clemmens et al. (2023, DOI: 10.1038/s41586-023-05746-w). If the prediction accuracy halves when going from 1 to .80, why would we even expect accuracy as high as .64 for age when the second measure, functional connectivity, likely has reliability between .2 - .5? How should we make sense of the interpretations given the interaction between phenotypic and brain reliabilities (i.e., Hedge et al discuss both measures' reliabilities)? Moreover, if the issue here is that the predictive accuracy decreases due to reliability, if the authors have a highly reliable measure (e.g., age = 1.0, grip strength = .93), wouldn't their point be strengthened if they demonstrate how these items have stable prediction *across* rsfMRI measurement occasions?

We absolutely agree with the reviewer that the reliability of both features and targets in multivariate predictions influences the final attenuation we observe. However, the focus of our work was on the reliability of behavioural phenotypes, because we believe that behaviour often falls by the wayside in large neuroimaging dataset initiatives. Furthermore, as pointed out by the reviewer, there are already previous studies pointing to the importance and impact of the reliability of fMRI features. Importantly, large dataset initiatives often do not follow established best practices from the psychometric literature to maximise the reliability of their phenotypic measurements (e.g. cognitive tasks are shortened). Moreover, various neuroimaging-derived features are used in the literature to predict behavioural traits and thus understanding how the reliability of targets impacts the results of these analyses is in and of itself useful. We also note that there is currently no way to translate Spearman's attenuation formula from the bivariate scenario to out-of-sample prediction accuracy of a model trained with regularisation on highly multidimensional features (a standard procedure in BWAS-type studies). Therefore, here we aimed to increase the community's awareness of the importance of phenotype selection, and provide some intuitions for researchers by empirically demonstrating the impact phenotypic reliability has on prediction analyses. In this framework, we chose the simpler scenario of illustrating this by only manipulating the phenotype reliability, while keeping the reliability of the brain features constant. Nevertheless, we recognise that our manuscript would benefit from a treatment of fMRI reliability and its impact on our results. We have therefore added a discussion of this as

already mentioned in our response to point 5 above. Importantly as mentioned in our response above, we recognise that poor rs-fMRI reliability worsens the attenuation of BWA associations/predictions and have therefore generalised some of our claims from “phenotypic reliability” to measurement reliability more broadly on lines 410-412:

“Although high reliability of either measurement is necessary for meaningful investigations of prediction accuracy, it is not sufficient.”

And on lines 355-358:

“Thus, it is important that on top of considering measurement reliability, researchers continue to follow guidelines for generalisable (Paus, 2010) and reproducible predictive modelling (Varoquaux, 2018; Janssen, Mourão-Miranda, & Schnack, 2018; Scheinost et al., 2019; Poldrack et al., 2020).”

In regards to questions of interpretation, we note that halving of prediction accuracy with the reduction of phenotypical reliability of 0.4 was observed in all datasets and behavioural phenotypes irrespective of prediction algorithms and analysis choices (main text figures 1 and 3, supplementary figures 3-6, 7, 8), even though the reliabilities of the functional connectivity measure differs across datasets (e.g., Marek and TC, 2022 mention reliability of connectivity in HCP as $r = 0.79$ and UKB as $r = 0.39$) as well as when we directly reduce the reliability of functional connectivity (new supplementary results 1,2 and 9 - see below). This suggests that irrespective of the dataset, increasing the reliability of behavioural assessments from a common ICC = ± 0.6 to 0.8 or higher can have a marked impact, even if the reliability of the fMRI data is not optimal to begin with. We have thus attempted to sharpen our claims in the discussion:

“Building on arguments emphasising the importance of reliability in biomarker research (e.g. Milham et al., 2021), we illustrate the amount of attenuation that can be expected by the reliability of routinely collected neuropsychological assessments available in many large datasets. Our results suggest that moderate reliability (ICC = 0.6 - 0.4) can produce serious attenuations of prediction accuracy irrespective of the dataset, rs-fMRI reliability and analytical choices. Moreover, even good levels of reliability (ICC = 0.6 - 0.8) were found to substantially attenuate brain-behaviour associations. In particular, reducing reliability to ICC = 0.6 diminished prediction accuracy on average by half.”

And (lines 304-312):

“... low phenotypic reliability that produced serious attenuation in the HCP-A dataset is likely to display even greater attenuation in datasets with less reliable fMRI measurements. ... However, we also emphasise that the impact of low phenotypic reliability generalised across datasets as well as when feature reliability was directly manipulated. Therefore, even with exceptionally reliable fMRI measurements, unreliable phenotypes are still likely to substantially attenuate out-of-sample prediction accuracy, as consistent ranking across individuals will be impaired.”

We have supported these arguments by adding three new supplementary simulation analyses showing 1) the attenuation of accuracy after reducing the reliability of functional connectivity across all phenotypes; and 2) as suggested by the reviewer, the attenuation of

prediction accuracies across fMRI occasions. Importantly, together these analyses indicate the rate of change in attenuation of prediction accuracy shows minimal variation across different rs-fMRI reliability. In-text references for these analyses are now on lines 134-6: **“Reducing the reliability of resting-state functional connectivity by shortening scan duration reduced the overall prediction accuracy, but did not impact the pattern of change in R2 (Supplementary Results Figure 5 and 6).”** and lines 157-160: **“Importantly, analysis choices such as confound regression, feature space or feature reliability resulted in small variations in prediction accuracy on empirical and simulated data but had no impact on the rate at which performance decreased (Supplementary Results Figure 7-9).”**

Supplementary Figure 5. Influence of feature reliability (functional connectivity) on age prediction with decreasing reliability. Impact of shortening rsfMRI timecourses from the average of all four sessions collected on both days to an average of 13 minutes collected on the first day only and finally connectivity calculated from a single 6 minutes session collected on the first day in the anterior-to-posterior direction (same as only session in the UKB).

Supplementary Figure 6. Influence of feature reliability (functional connectivity) on age prediction with decreasing reliability. Age prediction from day 1 and 2 rsfMRI acquisition separately. Both days had 13 minutes and 22 seconds of data.

Supplementary Figure 9. Influence of feature reliability (functional connectivity) on the prediction of total cognition with decreasing reliability. Impact of shortening rsfMRI timecourses from the average of all four sessions collected on both days to an average of 13 minutes collected on the first day only and finally connectivity calculated from a single 6 minutes session collected on the first day in the anterior-to-posterior direction (same as only session in the UKB).

7. The authors note that “highly reliable data in smaller samples can outperform large amounts of moderately reliable data”. I worry that this claim will be leveraged in service of underpowered correlation/prediction analyses. Most interesting psychological phenotypes are going to realistically have correlations with brain phenotypes that are relatively low (cf. Marek et al., 2022). Let’s say that we are studying a brain-behavior relationship where the true underlying correlation in the population is $r=0.2$ (which is well above the maximum correlations observed by Marek). A sample size of 193 is required for 80% power to find this association in a single test, assuming perfect reliability; of course, a researcher will usually want to look across multiple phenotypes, which will reduce power substantially due to multiple test corrections and thus require much larger samples. While I agree that higher reliability can improve power, I implore the authors to not provide researchers with an excuse to do badly powered studies.

We thank the reviewer for pointing out that our phrasing could be interpreted in a different way than we intended and potentially be used as an invitation for underpowered studies. As shown by Marek + TC (2022) power can be a serious issue in many traditional study designs, we mainly aimed to show that this ought not to be treated as a blanket statement across all behavioural phenotypes. While increasing reliability is not a silver bullet, it is worth pointing out that improving reliability will improve statistical power. We have now rephrased our conclusions in a more cautious manner and added a cautionary statement in the discussion:

“Here we demonstrate that depending on the true association strength, highly reliable phenotypes can reach comparable prediction accuracy using samples in the hundreds rather than thousands, as they are less subject to marked attenuation by low reliability. ... Finally, we note that our findings should not be taken to justify the use of small n studies under the guise of high measurement quality. As long as true associations between behavioural phenotypes and neuroimaging display small effect sizes, very large samples will be necessary. Thus, it is important that on top of considering measurement reliability, researchers continue to follow guidelines for generalisable (Paus, 2010) and reproducible predictive modelling (Varoquaux, 2018; Janssen, Mourão-Miranda, & Schnack, 2018; Scheinost et al., 2019; Poldrack et al., 2020).”

We also added a qualification to our results in lines 211-214 supported by new supplementary results:

“For phenotypes displaying weaker association with functional connectivity, this effect was less pronounced, requiring reliable training sets with half the size of the full sample with less reliable data to achieve comparable accuracy (Supplementary Results Figure 15).”

Supplementary Figure 15. Prediction and subsampling in UKB. (A) Impact of training set size on original and simulated cognitive flexibility with reduced reliability. Panel (B) Impact of training set size on prediction accuracy in empirical and simulated data with varying levels of reliability. Solid lines represent the mean across all 100 simulated datasets in each correlation band and shaded areas represent 2 standard deviations in prediction accuracies. Results were fitted with an exponential function for illustration purposes and were adjusted for the reliability of the TMT task ($ICC = 0.775$) estimated in independent data by Fawns-Ritchie et al. (2020).

8. In the discussions, the authors state: “Our results show that high reliability of phenotypes is paramount for the prediction of individual differences from neuroimaging” & “Excellent reliability of phenotypic assessments is paramount for investigating brain-behaviour associations.”. This may be a bit overstated for a couple of reasons. First, the evidence shows how model performance may be attenuated from the perspective of the outcome variable. Little investigation/discussion of rsfMRI measurement is provided. Furthermore, no repeated prediction across rsfMRI occasions is included. In light of low reliability in rsfMRI (see Noble et al., 2019), from “we observed that reliability levels lower than 0.6 can make the investigation of predictive validity (e.g. is functional connectivity a good predictor of phenotype X) meaningless” do the authors really think that that most, if not all, predictive utility analyses using rsfMRI will be “meaningless”. Second, to achieve “perfect” reliability may not always make sense. This relates to a point the authors noted in the introduction “This assumes the measure in question assesses a stable characteristic of the individual or the amount of change between occasions does not differ between individuals (e.g., due to practice from repeated testing).” Perfect reliability scores of function/structure are likely critical in certain circumstances, such as the use of fMRI for neurosurgical planning (as noted in Bennett & Miller, 2010, DOI: 10.1111/j.1749-6632.2010.05446.x). However, this may not always be reflective of the process/system being measured, which may fluctuate in the real world. For example, in developmental work reliabilities that are .90 or higher, in some cases, may be problematic. Why collect 3 waves of scores when they are extremely stable?

In fact, Cronbach's alphas that are .94+ are presumed to contain redundancies—often highlighting that the measure is oversaturated with items. So the reliability of the measure may depend on the context and research question(s). One wouldn't expect low within-subject variance on IQ scores and lack of shifts in between-subject variances as a function of environmental/developmental factors. This may instead suggest that the type of questions being asked and how the data are being modelled should be modified. Alternatively, without fluctuations in certain measures, characterizing changes in people across time would be a futile effort. Rather than writing off the entirety of a research program (e.g., all behavioural phenotypes are insufficient for analysis of rsfMRI associations), it would be helpful to contextualize the problem in units of measurement of different phenotypes and fMRI. Through this, perhaps the authors can offer some recommendations that researchers can use to navigate two highly unreliable measures?

We thank the reviews for their many insights. We address the multiple facets of this comment separately below

- 1) *Model performance attenuation from the perspective of the outcome variable without considering fMRI reliability: First of all, we agree with the reviewer that parts of our discussion may have come across as a bit overstated without considering fMRI reliability and have therefore addressed these concerns in our revised manuscript and discuss these in detail in points 5 and 6 above. As mentioned already, we now thoroughly discuss our results in relation to fMRI reliability as well as provide new supplementary results to support this. In summary, while low rsfMRI reliability attenuates prediction accuracy even further, it does not diminish the importance of improving behavioural phenotyping, especially as many measurements may have suboptimal test-retest reliability (Hedge et al., 2018; Scott, Sorrell, & Benitez, 2019; Enkavi et al., 2019; Taylor et al., 2020; Fawns-Ritchie & Deary, 2020; Anokhin et al., 2022). We do not think that all analyses of the predictive utility of rsfMRI for behaviour are going to be “meaningless”. We, however, do believe that given the current reliability of resting-state functional connectivity, predicting outcomes with low reliability will be in many cases difficult and will thus not be a fair evaluation of the predictive power of neuroimaging. We sincerely apologise that our framing of this issue was not clear enough and have therefore replaced it with **“Our results suggest that moderate reliability (ICC = 0.6 - 0.4) can produce serious attenuations of prediction accuracy irrespective of the dataset, rs-fMRI reliability and analytical choices.”** As noted above, our results in the HCP datasets may in fact represent the ‘best case scenario’ with high-quality neuroimaging data and acquisition times that are longer than in many other datasets. BWAS are therefore likely to be subject to even stronger attenuations in other datasets unless the sample size is many-fold larger as can be seen in our results in the UKB dataset (Figures 3 and 4).*
- 2) *To achieve “perfect” reliability may not always make sense: We agree with this point. Our aim here was not to suggest that researchers ought to strive for perfect reliability at all costs, but to point out the importance of well-designed behavioural measurements of individual differences by illustrating the impact of low phenotypic reliability on prediction accuracy, and to show that 1) out-of-sample prediction suffers from similar issues as does in sample correlation and 2) that very large samples will*

not compensate for the lack of precision in measuring behaviour. In our opinion, big data consortia have put too little focus on behavioural measurement quality, which negatively impacts our ability to associate fMRI recordings with behaviour. In light of this, we have given our argument more nuance to avoid being misunderstood and greatly expanded our section on the caveats of striving for perfect reliability alone which now reads:

“Although high reliability of either measurement is necessary for meaningful investigations of prediction accuracy, it is not sufficient. For instance, highly reliable phenotypes that don't capture a valid representation within the brain are not likely to improve effect sizes. Moreover, many behavioural measurements are validated against other established psychological scales or with specific populations in mind, rather than developed in light of their biological relevance. As a result, they may not be well-suited for investigations of brain behaviour associations, and thus, enhancing their reliability may bring little improvement in effect size. Similarly, structural MRI metrics that display better reliability than functional connectivity (Reuter, Schmansky, Rosas, & Fischl, 2012; Masouleh et al., 2020), are often poorer predictors of many psychological constructs (Ooi et al., 2022) that may instead rely on intrinsic fluctuations in neural activity (Waschke, Kloosterman, Obleser, & Garrett, 2021). Therefore, while optimising measurement reliability offers one possible avenue for improving the investigation of individual differences; it will not guarantee larger effect sizes (Finn & Rosenberg, 2021) or better prediction accuracy, especially if the selection of appropriate phenotypes is neglected.”

Finally, here we focused on test-retest reliability of trait-like phenotypic characteristics, which are considered to be relatively stable across time and situations, while it goes without saying that they (or their measurement) can be affected by interventions (e.g., treatment, practice, distraction). In our opinion, IQ is considered a rather stable trait and would thus not expect strong fluctuations, unless there are interventions/factors that lead to differential learning, distraction etc. and, thereby, to changes in test performance that differ between individuals. However, even in such cases, IQ can be measured with high reliability, as test-retest reliability and sensitivity to change are not mutually exclusive. High reliability does not entail stability over time in the absolute level of a trait. Consider a clinical assessment that is used to evaluate treatment success: it needs to be able to measure both the current level of symptomatology and its reduction over time. Thus, the more "short-lived" a certain trait's level of individual expression (because of interventions), the more care must be taken to assess a measure's test-retest reliability to prevent it from being confounded by true (and individually different amounts of) fluctuations in the trait of interest. Lastly, measurements of traits that vary over time in a way that is different across individuals (e.g. subjects with prodromal syndromes may decline faster) can still have excellent test-retest reliability over short time intervals and track this change accurately.

Based on the above discussion, we have rephrased our original claims, as follows:

“Our results show that high reliability of phenotypes is paramount for the prediction of individual differences from neuroimaging” now reads as **“Overall, these results indicate**

that high test-retest reliability of behavioural phenotypes is crucial to fairly evaluate the potential of neuroimaging for the prediction of individual differences in behaviour". "Excellent reliability of phenotypic assessments is paramount for investigating brain-behaviour associations." has been updated in the conclusion to summarise all our arguments from the above sections and now reads as:

"The recent availability of large-scale neuroimaging datasets, combined with advances in machine learning, has enabled the investigation of population-level brain-behaviour associations. In this study, we demonstrate that common levels of reliability across many behavioural phenotypes in such datasets can strongly attenuate or even conceal actual associations. This, in turn, can lead to scientifically questionable conclusions about the predictive potential of neuroimaging and hinders clinical translation. Therefore, greater emphasis needs to be placed on refining behavioural phenotyping in large datasets on top of similar efforts directed at neuroimaging. Together, more reliable neurobiological measurements and "markers" of behaviour will be necessary to fully exploit the benefits of big data initiatives in neuroscience, promote the identification of potential biomarkers, and contribute to reproducible science."

9. Regarding the statement "composite or summary scores are likely to lead to better prediction accuracy than individual assessments", this is not always true. Sum scores of multidimensional measures may alter/misrepresent effects and may not be an appropriate way to capture certain phenotypes (McNeish & Wolf, 2020, DOI: 10.3758/s13428-020-01398-0) because sum scores/averages impose a measurement model that suggests each loading and error is equal. In general, latent modelling frameworks, in the presence of multi-scale or multi-session development, can capture the latent construct in a manner that accounts for systematic/unsystematic error. How the latent value is derived depends on whether the assumption is that the construct exists in nature (e.g. reflective) or whether the latent construct depends on the scholar's construction/interpretation of the construct (e.g., formative; Coltman et al., 2008, DOI: 10.1016/j.jbusres.2008.01.013). That latent process can help with improved accuracy in representing the latent variable and improvement in its reliability, but if misspecified, may also misrepresent the scores like the sum score. Again here it would be helpful to express the measurement issues in a more sophisticated way.

We wholeheartedly agree with the reviewer and have disambiguated the discussion to make a distinction between simple summary scores and weighted latent models as well as pointing the readers to the relevant literature:

"In already acquired data, researchers should consider the reliability of measurements before embarking on machine learning analyses and select relevant variables with the best psychometric properties. For behavioural phenotypes, data reduction techniques such as principal component analysis or summary scores, assuming that error variance and loading of all items on a latent dimension are equal (McNeish & Wolf, 2020), can increase reliability and lead to larger effect sizes than

individual items (Lohmann et al., 2021; Tian & Zalesky, 2021; Marek et al., 2022; Ooi et al., 2022; Sasse et al., 2022). Supporting this, composite scores of the NIH toolbox tasks in the HCP datasets were more reliable than individual assessments and reached higher prediction accuracy... If equal item loading on a latent dimension cannot be assumed, reliability can be increased using latent modelling frameworks that account for systematic and unsystematic errors. However, more work is necessary to identify the most cost-effective strategies for optimising the reliability of both measurements without sacrificing measurement validity to increase prediction accuracy. To this end, future research should focus on ways to improve the reliability of already acquired data and evaluate best practices to preserve reliability when acquiring new data at large scales.”

Reviewer #2 (Remarks to the Author):

This is a timely investigation on an important topic of phenotypic measurement reliability in the brain-based multivariate prediction of psychological and health outcomes. The strengths of the study are clear: generally well-considered methodological choices, multiple independent datasets, and a cogent (and very welcomed) integration of psychometric theory and neuroimaging research. Strengths here, however, are not without some concerns and areas for further clarification. Most notably, clarifying that the methods but critically the results and implications here reflect a situation of making often low reliability phenotypes even less reliable (based on the simulation approach). Additionally, it would benefit the manuscript to include increased integration of psychometric research outside of neuroimaging (i.e., scale development) and a bit of information on the data generation model for simulation.

We thank the reviewer for the positive assessment of our work and the helpful feedback on our manuscript which assisted us in considerably improving it. We have clarified and extended our discussion of psychometric concepts, sharpened the conclusions we drew from our results and updated our results section to not make low reliability phenotypes appear less reliable than they are. Finally, we also included new supplementary analyses to further strengthen these changes. We elaborate on these points below.

1. In general, the authors appropriately state that the version of “reliability” here is not synonymous with reliability in the standard sense. However, some of these demarcations become very important with respect to the implications of the study and future recommendations. In general, the analyses reported herein focus on a “proof of principle” demonstration that added measurement noise decreases prediction accuracy, but the actionable information moving forward is less clear to me. In the simplest terms these results simulate adding noise and see what happens, ultimately, however, we as a field want to reduce noise and increase measurement reliability. That is the “best” highest reliability results presented here are actually the current state of brain-phenotype relations, progressively adding noise makes things worse but it’s not always clear under which circumstances this matters.

We absolutely agree that as a field we want to improve rather than reduce reliability. Our aim of using simulations to reduce the reliability of behavioural measurements was to offer general intuitions of the magnitude of attenuation one can expect when doing out-of-sample prediction by the reliability of behavioural measurements common in large datasets. With these simulations, we illustrate that variables with moderate reliability are likely subject to serious attenuation. Importantly, the highest reliability results we present are not necessarily the current state of brain-behaviour relations, given that we only selected variables with the

highest reliability ($ICC \approx 0.9$). Most variables that are currently used in prediction fall into the reliability range we simulated. This is supported by empirical data where we show the reliability of many behavioural phenotypes in current datasets (which is visible in supplementary results figures 12, 13 and now also summarised in additional supplementary figure 17, pasted below). Furthermore, prediction accuracy for such moderately reliable data was in the expected range based on our simulation results - with more reliable measures on average showing higher prediction accuracies (visible in Figure 2 in the main text and now additionally in the ABCD dataset in the new Supplementary figure 14 that was added to strengthen our findings of an association between phenotype reliability and its prediction accuracy) on lines 194-197 of the original manuscript: “Similar to our main analysis, all variables with reliability lower than < 0.6 displayed very low accuracy ($R^2 < 0.02$). Conversely, only variables with excellent reliability (the picture vocabulary task, total cognition, grip strength, reading English and crystallised cognition) could achieve $R^2 > 0.05$.”

Supplementary Figure 17. Boxplots of test-retest correlations (A) and ICC (B) of all measures used for assessing the association between prediction accuracy and reliability in Figure 2 of the main text. Overall median for test-retest correlations = 0.55; and ICC = 0.51. Both retest correlations and ICC are displayed as for some datasets, such as the ABCD, test-retest correlations may be more appropriate as systematic error coming from different rates of development across participants is not penalised.

Figure 2. Association between reliability and prediction accuracy. (A) HCP-YA and (B) UKB dataset. Each data point represents a behavioural assessment in each dataset.

Supplementary Figure 14. Association between reliability and prediction accuracy. Each data point represents one of 25 behavioural assessments in the ABCD. As measurements were collected from participants aged 9-10 at baseline and 11-12 at follow-up, reliability is calculated with test-retest (Pearson) correlation and not ICC. Unlike ICC, correlation is robust to systematic age-related changes as it is not penalised by differences in means between baseline and follow-up data (mean retest interval = 23 months) and different rates of development across participants.

Following the reviewer's comment, we revised our results and discussion to make clear that our simulations reflect reliabilities that are in a range that is representative of measurements available in large datasets specifically (Han & Adolphs, 2020; Taylor et al., 2020; Anokhin et

al., 2022) and assessments of cognition more generally (Enkavi et al., 2019; Hedge et al., 2018). Most notably in the results section (148-151):

“Next, we investigated the attenuation of prediction accuracy that can be expected in typical studies of brain-behaviour associations, by systematically adding noise to the most reliable measures (ICC \approx 0.9) available in the HCP-A dataset ($n = 550$, Supplementary Table 3). This way we simulated new phenotypes with reliabilities that are common in neuropsychological assessments and have plausible true effect sizes.”

The relevant section of the discussion now reads:

“The attenuation of a correlation between two variables by their reliability was already described by Charles Spearman in 1910. Here we aimed to demonstrate that machine learning approaches used to identify brain-behaviour associations also suffer from low phenotypic reliability and show its impact on out-of-sample prediction accuracy. Generally, we found that reliability attenuated out-of-sample prediction accuracy in a similar manner to what has been described for correlation (Nunnally, 1970; Vul et al., 2009; Zuo et al., 2019) and classification (Frenay & Verleysen, 2014; McNamara, Zisser, Beevers, & Shumake, 2022). Building on arguments emphasising the importance of reliability in biomarker research (e.g. Milham et al., 2021), we illustrate the amount of attenuation that can be expected by the reliability of routinely collected neuropsychological assessments available in many large datasets. Our results suggest that moderate reliability (ICC = 0.6 - 0.4) can produce serious attenuations of prediction accuracy irrespective of the dataset, rs-fMRI reliability and analytical choices. Moreover, even good levels of reliability (ICC = 0.6 - 0.8) were found to substantially attenuate brain-behaviour associations. In particular, reducing reliability to ICC = 0.6 diminished prediction accuracy on average by half.”

Finally, we have included additional supplementary analyses showing that improving reliability by e.g. compounding variables is still possible and that this also improves prediction accuracy. We also highlight other results where this is visible (lines 398-405):

“Supporting this, composite scores of the NIH toolbox tasks in the HCP datasets were more reliable than individual assessments and reached higher prediction accuracy. Similarly, averaging left and right hand grip strength in the UKB dataset compared to each hand separately lead to improvement in both reliability and accuracy. Comparable increases were also achieved when grip strength was averaged across testing occasions (Supplementary Results Figure 18).“

Supplementary Figure 18. Impact of increasing reliability by averaging in UKB hand grip strength prediction. Numbers denote UKB time points. Neuroimaging was collected at time point 2. Grip strength was averaged over time points 2 (neuroimaging baseline) and 3 (neuroimaging follow-up) to maximise the number of subjects. Reliability was calculated as Pearson test-retest correlation between time point 0 (baseline) and time point 2 as well as the average of 2 and 3. HGS_lr stands for the average of measurement over left and right hands. Abbreviations; HGS: Hand grip strength.

In more detail, consider the case of $r(\text{original}, \text{simulated}) = .8$, if the true phenotypic reliability is .5, it's a bit unclear to me how to interpret the magnitude of what $r(\text{orig}, \text{sim})$ of .8 really means here in absolute terms with regard to actionable information on the phenotypic reliability. Can the authors clarify this? At minimum I agree with the authors rationale, that it would seem that simulated vector with $r(\text{orig}, \text{sim}) = .8$ for a given measure with an existing phenotypic reliability of .5 would index reliability of even lower than .5 of course (as an extra noise process is being added). Yet, while the authors mention this "Thus, the 'true' reliability of simulated phenotypes is lower than the preset level of correlation between simulated and original data, as this preset level will be attenuated by each variable's original reliability.", it is unclear how these magnitudes influence absolute judgements and inferences and general recommendations based on the reported results. For example, on the age variable (which the authors rightfully point out is likely most consistent with reliability as age is considered measured with nearly zero error), one can draw a general recommendation that reliability matters (added noise decreases predictive accuracy). But the influences on other phenotypes (e.g. Fig 1 Cryst Cog, Grip Strength, Total Cog) with presumably a range of general "to-be-predicted effect sizes" is less clear given this magnitude interpretation challenge. For example $r(\text{orig}, \text{simulated}) = 1$ in these plots represents the measures with the current (less than ideal reliabilities they have), progressively adding noise here makes them

worse yes, but the authors also mention that we should be cautious of our current measures. I think a full simulation-based approach that disaggregates the “measures at hand” and whatever range of ~multivariate predictive effect sizes therein from the issues of reliability would provide a much stronger base to make recommendations. That being said, I appreciate the complexity of this given the high dimensionality of the imaging data and the predictive algorithms. In general, further clarifying the issue of “adding noise” to an already questionably reliable measure, the range of possible ~multivariate predictive effect sizes, and limitations on what recommendations can be concretely made moving forward (recognizing that many of these simulations represent making low-reliability measures even worse) is necessary.

We would like to thank the reviewer for pointing out the ambiguity in parts of our results presentation. Our reasoning was two-fold: Firstly, we presented $r(\text{original, simulated})$ because we created our simulated variables to specifically preserve the correlation between original and simulated data. Secondly, we didn't want to settle for a specific phenotypic reliability. The HCP-Aging (and many other) datasets don't provide retest data, and phenotypic reliability varies for many phenotypes across publications and populations (e.g. total cognition composite ICC = 0.86 - 0.95; Akshoomoff et al. (2013); Heaton et al. (2014)). However, we agree that this design choice negatively impacted the interpretation of our results and have therefore adjusted our figures and results accordingly. That is, in the revised manuscript we now present the simulated test-retest reliability rather than $r(\text{original, simulated})$ for all our results. Furthermore, we also correct our figures by the phenotypical reliability of specific variables (ICC_{total cog.} = 0.9 Heaton et al. (2014); ICC_{crystalized cog.} = 0.86 Heaton et al. (2014); ICC_{grip strength} = 0.93 Reuben et al. (2013) for HCP- Aging and by 0.84 for grip strength (Bohanon 2014) in UKB). For completion, we also include the uncorrected results in the supplementary results Figure 11.

In regard to recommendations for moving forward, we have attempted to sharpen our interpretation and the consequences of our findings as mentioned in our response to point 1 above. We also now provide a more detailed discussion of ways to improve the reliability of both existing datasets and to-be-collected datasets in lines 387-394:

“Prior to acquisition, this can be achieved by opting for a deeper phenotyping design (Gratton, Nelson, & Gordon, 2022) either in the laboratory or by means of ecological momentary assessment (Moskowitz & Young, 2006), introducing more rigorous testing strategies such as collecting more trials (for an overview see Zorowitz & Niv, 2022), taking measures to increase between-subject variance (Xu et al., 2023) or acquiring multiple assessments for data aggregation (Nikolaidis et al., 2022).”

“In already acquired data, researchers should consider the reliability of measurements before embarking on machine learning analyses and select relevant variables with the best psychometric properties. For behavioural phenotypes, data reduction techniques such as principal component analysis or summary scores, assuming that error variance and loading of all items on a latent dimension are equal (McNeish & Wolf, 2020), can increase reliability and lead to larger effect sizes than individual items (Lohmann et al., 2021; Tian & Zalesky, 2021; Marek et al., 2022; Ooi et al., 2022; Sasse et al., 2022). Supporting this, composite scores of the NIH toolbox

tasks in the HCP datasets were more reliable than individual assessments and reached higher prediction accuracy... If equal item loading on a latent dimension cannot be assumed, reliability can be increased using latent modelling frameworks that account for systematic and unsystematic errors. However, more work is necessary to identify the most cost-effective strategies for optimising the reliability of both measurements without sacrificing measurement validity to increase prediction accuracy. To this end, future research should focus on ways to improve the reliability of already acquired data and evaluate best practices to preserve reliability when acquiring new data at large scales.

2. The authors do well to be clear on the goal of “relative reliability” for phenotypic selection in the current work, but I would encourage a further consideration of general psychometric work. For example, in many ways at the stage of phenotypic selection in large-scale existing datasets, “the die has been cast” regarding phenotypic measurement selection and by extension the associated psychometric properties therein. That is, that, as the authors partially allude to but I would consider ways to further increase this, individual phenotypic measures via scale development are often designed or “validated” with psychometric targets in mind. There are of course incentives at play in scale development and I appreciate the authors alluding to distinctions between reliabilities reported in scale development and those observed in real large-scale data, but I would encourage the authors to consider how the individual measures themselves are developed and what we should do with “end of the pipeline use cases” (relative selection of phenotypes). I can imagine more concrete recommendations being that 1. We need to improve measurement reliability generally (although its relative impact on minimal sample sizes isn’t entirely clear see point 1 above) and 2. When looking at large-scale data we should emphasize selecting the most reliable phenotypic measures (the inferences here are based on the relative selection of phenotypes). Both general ideas appear in the manuscript but deserve further emphasis, particularly in the discussion section.

We appreciate the reviewer’s suggestion and expanded our discussion of actionable recommendations for researchers given our findings and previous literature on the topic. Furthermore, we have now split our discussion into implications and recommendations for 1) researchers planning data collection and 2) cases where data is already acquired and already indicated the relevant sections in the point above.

In regards to the validation of original measurements, we agree that this is a critical issue as measurements are often validated mainly against other established scales, rather than against methodologically different measures of the same construct (in particular not: neuroscientific measures), which can lead to an overestimation of their validity due to shared method-related variance, which in turn may lead to “surprises” of meagre association strengths in BWAS type studies. However, we believe that an extensive discussion of this topic will take the focus away from the main message of our manuscript and have therefore included it only briefly:

“Although high reliability of either measurement is necessary for meaningful investigations of prediction accuracy, it is not sufficient. For instance, highly reliable phenotypes that don’t capture a valid representation within the brain are not likely to improve effect sizes. Moreover, many behavioural measurements are validated against other established psychological scales or with specific populations in mind, rather than developed in light of their biological relevance. As a result, they may not be well-suited for investigations of brain behaviour associations.”

3. Can the authors provide a bit more information on the data generation model for the simulated data? I can imagine solutions for such a data generation model that creates vectors for a given correlation with the original, but it would be useful to include for example relevant formulae/proofs or more detail on the procedure.

We thank the reviewer for pointing out the missing information for our data generation model. We have included this information as well as an R implementation of all our simulated datasets online:

“In order to induce increasing levels of noise in the target variable, we created datasets that correlated with the originally observed (empirical) targets at a pre-specified Pearson’s correlation. This method was chosen to increase the interpretability of the resulting attenuation of brain-behaviour associations by controlling the amount of noise. The data generation procedure was as follows: First, a random vector was sampled from a standard normal distribution with the same mean and standard deviation as the original empirically acquired data (in the HCP these were age-adjusted and normalised to mean = 100 and SD = 15). Next, we calculated the residuals of a least squares regression of the sampled vector (X) on the empirical data (Y). The resulting orthogonal vector representing the portion of X that is independent of Y was then again combined with the original empirical data Y through scaling by the pre-specified correlation. This adjustment process manipulated the relative contributions of Y and the residuals of X on Y in the resulting simulated vector. The formula used for this process was:

$$X_{Y\rho} = \rho \sigma(Y \perp)Y + \sqrt{1 - \rho^2} \sigma(Y)Y \perp$$

Where $X_{Y\rho}$ is the new ‘simulated’ vector that correlates with the empirical data Y at a predefined correlation ρ . Y represents the residuals of a least squares regression of a randomly sampled vector X against Y. We provide an R implementation in the accompanying repository online (<https://github.com/MartinGell/Reliability/code>).“

Reviewer #3 (Remarks to the Author):

In the manuscript, "The Burden of Reliability: How Measurement Noise Limits Brain-Behaviour Predictions", Gell and colleagues demonstrate how imperfect reliability of behavioural phenotypes can result in reduced associations between the brain and behaviour. In-and-of-itself this is not a novel concept (attenuation of correlation [Spearman, 1904]), but the authors did a nice job of demonstrating how this phenomenon plays out in simulation and real data from the HCP and UK Biobank datasets. I think their main argument that "only highly reliable data can fully benefit from increasing sample sizes from hundreds to thousands of participants" was well supported. Thus, the paper was methodologically well done. However, my excitement around this topic was dampened for several reasons. For one, improved measurement reliability was highlighted in the Marek + TC BWAS paper as an avenue needed for more reproducible science, so I am not sure of the novelty of this paper beyond that. There are also several other papers that have come out over the last year showing the same (e.g., Nikolaidis et al., 2022; Tiego & Fornito, 2022). The specific argument that high reliability is a prerequisite for robust brain-behaviour relationships is likewise not novel (Vul et al., 2009; Milham et al., 2021).

We thank the reviewer for taking the time to assess our work and the valuable feedback which helped us to improve our manuscript.

In regard to novelty beyond Marek + TC (2022) we would like to point out that the aforementioned paper actually claims that increasing reliability will unlikely increase effect size due to "fundamental biological limits on the strength of the true association and/or the limitations of behavioural phenotyping and MRI physics" (Marek et al., 2022) and thus suggest increasing reliability will have only minor impact on reproducibility of science, if any. Furthermore, they base their assumption on rather high reliabilities reported in the manual of the NIH toolbox, while our results (and those by others, e.g.; Anokhin et al., 2022) demonstrate that reliabilities "in the wild" are likely to be lower. Here we aimed to illustrate that this statement in Marek et al., (2022) cannot be taken as a blanket statement across most measurements of behaviour and point out there is still much room for improvement. Many assessments collected across datasets display moderate to good reliability (Hedge, Powell, & Sumner, 2018; Enkavi et al., 2019; Fawns-Ritchie & Deary, 2020; Anokhin et al., 2022) and thus the degree of BWAS attenuation in out-of-sample predictions the field can expect is at this time unclear.

In regards to papers putting forth similar arguments, we would like to emphasize that while much previous work provides key arguments for the importance of reliability (Vul et al., 2009; Milham et al., 2021; Nikolaidis et al., 2022; Tiego & Fornito, 2022), empirical work on this topic is currently very sparse, leaving the impact this has on the current state of the art in BWAS unknown. More importantly, none of this work provides intuitions on how reliability attenuates out-of-sample prediction (but rather on correlations) which is the current state of the art in BWAS. We thus aim to provide the community with empirical data that show the extent of the attenuation on data that researchers are used to working with (i.e. rsfMRI and behavioural assessments) in the datasets they are familiar with. There is no simple way to

translate our understanding of correlation attenuation discussed in previous work to out-of-sample prediction, thus warranting its own investigation. To address these topics we now provided a more focused treatment of the impact of behavioural reliability on out-of-sample prediction. These points also provide evidence for the contribution of the current work over and above what was identified by Spearman in 1904. Finally, while association attenuation is not a novel concept, it is still not widely appreciated as is exemplified by the recent theoretical work cited above. We therefore hope our empirical work on the topic will provide convincing evidence.

1. The authors mention in the Discussion that "Large samples are necessary but not sufficient" for improved brain-behaviour reproducibility. However, within that section, the authors appear to contradict themselves by saying "Here we demonstrate that reliable phenotypes may be predicted using samples in the hundreds rather than thousands, as highly reliable measurements are not subject to attenuation by reliability". The latter clause is a truism, and in the first clause, the authors mistakenly imply that a measure with high reliability (e.g., shoe size) will ensure a large enough effect size with the brain to detect it with a small sample size. This may be true for age, but it is certainly not true for all non-brain phenotypes.

*We thank the reviewer for their insightful comments. We completely agree that high reliability will not ensure a large enough effect size that is detectable with a small sample size and did not mean to imply this. We apologize that our wording led to such a reading. In fact, by the section heading "Large samples are necessary but not sufficient" we meant exactly that the investigation of the effect sizes common in brain-behaviour associations need both large N and sufficient reliability, which is illustrated by our results from the UKB. The second claim describing a simple truth that the higher the attenuation by reliability, the lower the effect size and thus the larger the sample size needed to identify it, was not meant to contradict this. We simply aimed to illustrate empirically, in data that researchers in the field are familiar with and methods they use (i.e. out-of-sample prediction accuracy), that comparable prediction effect sizes can be detected in hundreds of subjects with highly reliable data as in thousands of subjects with unreliable data. We have therefore rephrased our claims to state this more clearly as **"Here we demonstrate that depending on the true association strength, highly reliable phenotypes can reach comparable prediction accuracy using samples in the hundreds rather than thousands, as they are less subject to marked attenuation by low reliability."** and have rephrased our results (lines 209-212) to reflect our arguments better:*

"Importantly, a change of 0.2 in reliability had a larger impact on prediction performance than a change in training set size. For age prediction, even samples of 652 subjects with excellent reliability ($r = 0.81$; $R2_{\text{mean}} = 0.15$, $R2_{\text{sd}} = 0.005$) produced comparable accuracy to the full sample ($n = 4450$) with a moderate level of phenotypic reliability that is common across behavioural assessments ($r = 0.49$; $R2_{\text{mean}} = 0.14$, $R2_{\text{sd}} = 0.01$)."

Secondly, and more importantly, we believe that what these results show is that highly reliable data are necessary in combination with large samples to fully investigate the predictive power of neuroimaging, as the strength of “true associations” is hard to infer. This is because Spearman’s disattenuation formula does not translate to the out-of-sample prediction accuracy of a model trained on multidimensional brain features with regularisation based on data-driven hyperparameter tuning. Thirdly, we have included new supplementary results using a behavioural measure expected to, in contrast to age, have a lower association with functional connectivity. These new analyses show that, while still benefitting from increasing reliability, smaller effect sizes still need thousands of subjects to be predictable even when reliability is high. This is illustrated in supplementary figure 15 and included in the results (lines 212-215):

“For phenotypes displaying weaker association with functional connectivity, this effect was less pronounced, requiring reliable training sets with half the size of the full sample with less reliable data to achieve comparable accuracy (Supplementary Results Figure 15).”

Supplementary Figure 15. Prediction and subsampling in UKB. (A) Impact of training set size on original and simulated cognitive flexibility with reduced reliability. Panel (B) Impact of training set size on prediction accuracy in empirical and simulated data with varying levels of reliability. Solid lines represent the mean across all 100 simulated datasets in each correlation band and shaded areas represent 2 standard deviations in prediction accuracies. Results were fitted with an exponential function for illustration purposes and were adjusted for the reliability of the TMT task (ICC = 0.775) estimated in independent data by Fawns-Ritchie et al. (2020).

Finally, we absolutely agree with the reviewer that high reliability will not ensure a large enough effect size with the brain to detect it with a small sample size. We have therefore expanded our discussion of the “shoe size fallacy” and discuss this in more detail in our response to point 3 below.

2. Although I agree 100% that improved reliability of behavioural measures is needed, the elephant in the room - even in this paper - is clearly the sample size. One could argue from these results that both reliability and thousands of individuals are needed for improved reproducibility in BWAS studies. For example, in Fig. 3a (prediction of age), a training sample size of ~400 results in a prediction accuracy of ~0.10, whereas a sample of ~4,400 boosts it to ~0.40. As the authors show, reliability and prediction accuracy are tightly linked, but so are sample size and prediction accuracy. To me, this is overwhelming evidence that we need both a large sample and reliability. This needs to come through more clearly in the text.

We completely agree with the reviewer that both large samples and high reliability are needed for reproducible BWAS. The importance of sample size per se has been shown previously and thus was not the focus of our work. We aimed to point out that continuously increasing sample sizes when the variables of interest are measured with suboptimal reliability will be of limited benefit for the field - where cross-sectional associations of brain and behaviour are concerned. Instead, the field should strive to improve measurement practices before increasing sample sizes much further by demonstrating that increasing reliability can be a more fruitful endeavour (Figures 3 and 4 of the main text). That said, we also agree (and show) that large sample sizes do matter and acknowledge that we did not explicitly say this in the original version of the manuscript. As already mentioned in the response to concern 1, we have greatly expanded our discussion to highlight the need for large samples, over and above increasing the reliability of measurements, even if this also increases effect size. The whole section now reads:

“Here we demonstrate that depending on the true association strength, highly reliable phenotypes can reach comparable prediction accuracy using samples in the hundreds rather than thousands, as they are less subject to marked attenuation by low reliability. These results suggest that collecting more reliable data may be particularly important for research questions where, assuming cross-sectional design is appropriate, many thousands of participants are difficult to acquire (e.g., specific conditions) and discuss ways to implement this below. However, more importantly, we demonstrate that only reliable phenotypes can fully benefit from improvements in prediction accuracy as training set sizes increase from hundreds to thousands of participants that has been observed previously (Nieuwenhuis et al., 2012; Jollans et al., 2019; Traut et al., 2022). Conversely, measurements with poor reliability are likely suboptimal candidates for big data initiatives as collecting thousands of participants will only yield minor increases in accuracy before saturating. Therefore, improving the measurement reliability of appropriately selected phenotypes for associations with neuroimaging features will likely boost predictive (and statistical) power in large datasets. Finally, we note that our findings should not be taken to justify the use of small n studies under the guise of high measurement quality. As long as true associations between behavioural phenotypes and neuroimaging display small effect sizes, very large samples will be necessary. Thus, it is important that on top of considering measurement reliability, researchers continue to follow guidelines for generalisable (Paus, 2010) and reproducible predictive modelling (Varoquaux, 2018;

Janssen, Mourão-Miranda, & Schnack, 2018; Scheinost et al., 2019; Poldrack et al., 2020).”

3. Difficult to acquire data. The authors note, "Particularly for research questions where thousands of participants are difficult to acquire (e.g., specific conditions), optimising for greater measurement reliability may be of greater benefit than increasing sample size. Here again, the authors are "either-or" in their thinking. What if the measure is reliable, but not a valid construct in the brain? Perhaps if data are difficult to acquire for a given population, a different research design is needed other than cross-sectional brain-behaviour correlations?"

*We thank the reviewer for their insightful comments and, as hopefully already addressed by our responses to points 1. and 2., we completely agree that there is no either-or option, but rather that both reliability and sample size matter. The reasoning for our statement was, that given a valid reason for a cross-sectional design, many researchers find themselves in a scenario with a limited budget or the population of interest being difficult to recruit (for example patients). In such a case, opting for a more reliable measurement strategy such as a deeper phenotyping approach may be a more feasible or cost-effective option. We have attempted to clarify our statement in the discussion on lines 341-344 to make this more explicit: **“These results suggest that collecting more reliable data may be particularly important for research questions where, assuming cross-sectional design is appropriate, many thousands of participants are difficult to acquire (e.g., specific conditions) and discuss ways to implement this below.”** As already evident from the quote, we have now also included a section discussing actionable recommendations for improving reliability before data collection:*

“Prior to acquisition, this can be achieved by opting for a deeper phenotyping design (Gratton, Nelson, & Gordon, 2022) either in the laboratory or by means of ecological momentary assessment (Moskowitz & Young, 2006), introducing more rigorous testing strategies such as collecting more trials (for an overview see Zorowitz & Niv, 2022), taking measures to increase between-subject variance (Xu et al., 2023) or acquiring multiple assessments for data aggregation (Nikolaidis et al., 2022).”

Finally, we agree with the reviewer that the validity of any measurement for the use in cross-sectional BWAS is an important issue and have greatly expanded our discussion on the appropriate selection of phenotypes and caution the readers against the shoe size fallacy by:

“Although high reliability of either measurement is necessary for meaningful investigations of prediction accuracy, it is not sufficient. For instance, highly reliable phenotypes that don’t capture a valid representation within the brain are not likely to improve effect sizes. Moreover, many behavioural measurements are validated against other established psychological scales or with specific populations in mind, rather than developed in light of their biological relevance. As a result, they may not be well-suited for investigations of brain behaviour associations. Similarly, structural MRI metrics that display better reliability than functional connectivity (Reuter, Schmansky, Rosas, & Fischl, 2012; Masouleh et al., 2020), are often poorer predictors

of many psychological constructs (Ooi et al., 2022) that may instead rely on intrinsic fluctuations in neural activity (Waschke, Kloosterman, Obleser, & Garrett, 2021). Therefore, optimising measurement reliability offers one possible avenue for improving the investigation of individual differences; it will not guarantee larger effect sizes (Finn & Rosenberg, 2021) or better prediction accuracy if the phenotype selection is neglected.”

4. As the authors note, achieving excellent reliability (>0.9) in large consortia studies has proven difficult (5/56 across HCP and UKB in the current paper). This is not even touching on the brain data, especially resting state fMRI which certainly has poor reliability within the UKB (6 mins of data). What do the authors think this implies for existing consortia given the joint reliability of the existing brain-behaviour data must be suboptimal (i.e., less than excellent)?

This is an important point that has also been raised by reviewer 1. Initially, we aimed to keep the manuscript focused on the reliability of behaviour, but we agree that this simplification made our manuscript less convincing. We now provide a discussion of our results in light of fMRI reliability and the differences in the reliability of neuroimaging between the datasets we use. Additionally, we have added supplementary analyses where the reliability of functional connectivity was directly manipulated. Of note, our simulation results held across all datasets that were used irrespective of the reliability of the rsfMRI measurements. Relevant passages now state:

“The final attenuation of brain-behaviour relationships will be determined by the joint reliability of both neuroimaging features and behavioural targets (Nunnally, 1970; Nikolaidis et al., 2022). Reliability of functional connectivity depends on the network (Tozzi, Fleming, Taylor, Raterink, & Williams, 2020), preprocessing steps (Noble et al., 2019) and scan duration, with longer acquisition leading to greater reliability (Noble et al., 2017, 2019; Cho, Korchmaros, Vogelstein, Milham, & Xu, 2021). The marked difference in age prediction accuracy between HCP and UKB datasets we observed here, is therefore, likely related to differences in rsfMRI acquisition (6 minutes in UKB compared to 26 minutes in HCP-A), in addition to lower precision in reported age in the UKB (measured in years compared to months in HCP). In other words, low phenotypic reliability that produced serious attenuation in the HCP-A dataset is likely to display even greater attenuation in datasets with less reliable fMRI measurements. Therefore, the results shown here may represent an optimistic scenario for the field, as 26 minutes of resting-state images collected over two days is, especially in clinical settings, uncommon. However, we also emphasise that the impact of low phenotypic reliability generalised across datasets as well as when feature reliability was directly manipulated. Therefore, even with exceptionally reliable fMRI measurements, unreliable phenotypes are still likely to substantially attenuate out-of-sample prediction accuracy, as consistent ranking across individuals will be impaired.”

Supplementary Figure 5. Influence of feature reliability (functional connectivity) on age prediction with decreasing reliability. Impact of shortening rsfMRI timecourses to 13 (retest correlation between day 1 and 2 = 0.55) and 6 minutes ($r = 0.46$) on first rsfMRI session.

Supplementary Figure 9. Prediction of selected phenotypes with feature-wise confound (age, sex) regression. Grip strength could not be predicted when confounds were regressed.

In terms of existing consortia, as noted in the passage above, datasets with poor rs-fMRI measures will experience even greater attenuation which together with poor behavioural reliability may even in certain cases lead to effect sizes close to zero. To help mitigate this in existing datasets we have now included recommendations for researchers:

“In already acquired data, researchers should consider the reliability of measurements before embarking on machine learning analyses and select relevant variables with the best psychometric properties. For behavioural phenotypes, data reduction techniques such as principal component analysis or summary scores, assuming that error variance and loading of all items on a latent dimension are equal

(McNeish & Wolf, 2020), can increase reliability and lead to larger effect sizes than individual items (Lohmann et al., 2021; Tian & Zalesky, 2021; Marek et al., 2022; Ooi et al., 2022; Sasse et al., 2022). Supporting this, composite scores of the NIH toolbox tasks in the HCP datasets were more reliable than individual assessments and reached higher prediction accuracy. Similarly, averaging left and right hand grip strength in the UKB dataset compared to each hand separately lead to improvement in both reliability and accuracy. Comparable increases were also achieved when grip strength was averaged across testing occasions (Supplementary Results Figure 18). If equal item loading on a latent dimension cannot be assumed, reliability can be increased using latent modelling frameworks that account for systematic and unsystematic errors. However, more work is necessary to identify the most cost-effective strategies for optimising the reliability of both measurements without sacrificing measurement validity to increase prediction accuracy. To this end, future research should focus on ways to improve the reliability of already acquired data and evaluate best practices to preserve reliability when acquiring new data at large scales.”

Supplementary Figure 18. Impact of increasing reliability by averaging in UKB hand grip strength prediction. Numbers denote UKB time points. Neuroimaging was collected at time point 2. Grip strength was averaged over time points 2 (neuroimaging baseline) and 3 (neuroimaging follow-up) to maximise the number of subjects. Reliability was calculated as Pearson test-retest correlation between time point 0 (baseline) and time point 2 as well as the average of 2 and 3. HGS_lr stands for the average of measurement over left and right hands. Abbreviations; HGS: Hand grip strength.

Reviewer #4 (Remarks to the Author):

In this manuscript, the authors show how the reliability of measured phenotypes (i.e., demographics, behaviour) impacts the predictive accuracy of machine learning models trained on functional connectivity data. They show that, as reliability diminishes, so too does predictive accuracy. Importantly, they also show that, past certain levels of measurement noise, it is more effective to invest efforts into improving phenotypic reliability than to simply collect more data. This is a timely and well-written manuscript of particular relevance in an era of large-scale neuroimaging data collection. I have no qualms with the take-home message of the manuscript, which is long-established if under-appreciated (e.g., see Bollen (1989), for a lengthy treatment of reliability on R-squared in multiple regression). I do, however, have outstanding questions that I hope the authors might consider addressing in the Discussion.

We thank the reviewer for this positive assessment of our work and the helpful feedback on our manuscript.

Major points

1. The observed rate at which predictive accuracy diminishes with reliability is striking. For example, in the manuscript, the authors report that $r^2 = 0.64$ of variance in age can be explained by functional connectivity. When the reliability of age is artificially diminished to 0.80, the prediction accuracy is halved to $r^2 = 0.33$. It is worth putting this in the context of Spearman's attenuation formula (Spearman, 1910): $r_{\text{obs}} = r_{\text{true}} * \sqrt{r_{\text{xx}} * r_{\text{yy}}}$, where r_{xx} and r_{yy} are the reliabilities of two variables, x and y ; r_{true} is their true (latent) correlation; and r_{obs} is the observed correlation.

For argument's sake, let us assume that the functional connectivity composite has a reliability of $r_{\text{xx}} = 0.9$ and the true correlation between this composite and age is $r_{\text{xy}} = 0.9$. Plugging these numbers in, we get: $0.9 * \sqrt{0.9 * 1.0} = 0.81$ ($r^2 = 0.65$), or about what is reported above. Now, plugging in the same numbers when the reliability of age is artificially decreased, we get: $0.9 * \sqrt{0.9 * 0.8} = 0.76$ ($r^2 = 0.58$). The variance explained in this toy example is substantially higher than what was actually observed ($r^2 = 0.58$ vs. 0.33). This begs the question as to why.

One possibility is that the multiple regression case is more complex than the bivariate case (e.g., due to interactions between imperfectly measured predictors; Bollen, 1989). A related possibility is that this reflects something specific to functional connectivity data (e.g., correlated measurement error). A third possibility is that this reflects something specific to the method (e.g., machine learning models that capitalize on chance during training perform even worse during out-of-sample prediction).

It would be useful for the authors to speculate why it is the case that even phenotypes with conventionally acceptable reliability (i.e., $\rho = 0.8$) lead to such dramatic decreases in predictive accuracy. (It would be even more helpful for the authors to demonstrate this via additional simulations, perhaps using fully simulated datasets where they can control all relevant features of the data). Such discussion might spur new research into alternative approaches to handling "suboptimal" reliability, which may prove equally fruitful as the likely substantial effort required to "acceptably" reliable behavioural measures (i.e., $\rho = 0.8$) to excellent levels of reliability (i.e., $\rho \geq 0.9$).

We thank the reviewer for the example illustrating that for simple correlation we would expect less decrease in prediction accuracy with decreasing reliability than we observed for out-of-sample prediction accuracy. Firstly, R^2 (coefficient of determination) in the machine learning framework is based on a sums-of-squares formulation rather than on squaring the correlation coefficient (Poldrack, Scheinost). Here, R^2 is calculated as (source: https://scikit-learn.org/stable/modules/model_evaluation.html#r2-score-the-coefficient-of-determination):

$$R^2(y, \hat{y}) = 1 - \frac{\sum_{i=1}^n (y_i - \hat{y}_i)^2}{\sum_{i=1}^n (y_i - \bar{y})^2}$$

Where \hat{y}_i is the predicted value of the i -th sample and y_i is the corresponding true value for total n samples. \bar{y} Represents the mean across all y . Therefore, if a model makes perfect predictions, its associated R^2 value will be 1.0, while a model making random predictions should have an R^2 value of approximately 0. Importantly, if a model's predictions are less accurate than they would be if the model simply returned the mean value for the data set (across y), the R^2 will be negative. This formulation therefore entails that the coefficient of determination can and often will have values different to the correlation squared (e.g. see Scheinost section 4.2 for an illustration of this). To clarify these issues we have focused our arguments by highlighting out-of-sample prediction throughout the manuscript. We have also now updated the calculation of our accuracy metrics in the methods section at lines 654-660:

" R^2 represents the proportion of variance (in the target variable) that has been explained by the independent variables in the model and is calculated here as

$$R^2(y, \hat{y}) = 1 - \frac{\sum_{i=1}^n (y_i - \hat{y}_i)^2}{\sum_{i=1}^n (y_i - \bar{y})^2}$$

Where \hat{y}_i is the predicted value of the i -th sample and y_i is the corresponding true value for total n samples. \bar{Y} Represents the mean across all y . In this formulation, the R^2 is not interchangeable with correlation coefficient squared."

Furthermore, given the way less reliable datasets were simulated in this study by creating correlated datasets with the original data, the attenuation formula would have to be adjusted to: $r_{obs} = r_{true} * \sqrt{r_{xx}} * r_y$, where r_{xx} stands for the reliability of rsfMRI connectivity and r_y stands for the reliability of simulated data. This is because by creating each less reliable dataset we only create a single 'test' (y) measurement. To create a true 'test-retest' dataset, the correlation/ICC between two such simulated less reliable datasets (i.e. $correlation(y_1, y_2)$) would have to be preset, rather than the correlation between the original participant data and simulated datasets (i.e. $correlation(Y, y_n)$). The relationship between these two correlations is thus $correlation(y_1, y_2) = \sqrt{correlation(Y, y)}$. We have therefore adjusted our results to reflect test-retest reliability in such a way.

Finally, there is no simple way to translate our understanding of bivariate correlation (or correlation squared) attenuation to out-of-sample accuracy in multivariate prediction. In the prediction case, the attenuation is not strictly driven by the reliability of predictors and targets in the way that is calculated by Spearman ($\sqrt{r_{xx} * r_{yy}}$) but the reliability of function or mapping $f(x, y)$ that was learnt by the model as we investigate the attenuation of the functions predictions. Thus, the reason for the discrepancy between our results and what would be expected given the calculations kindly provided by the reviewer is likely a combination of all the possibilities listed above. That is, the large feature space (79800 predictor variables) used for predictions is substantially more complicated than the bivariate association between a predictor and target. This is further complicated by the nature of features which are computed from fMRI measurements that are subject to severe spatial autocorrelation. Finally, the attenuation is studied here in a metric that is computed based on predictions made on a test set (i.e. out-of-sample), resulting from a mapping learnt by the algorithm, and not on the training set as is commonly done in the linear regression case.

Minor points

2. It should be noted that one of the author's main points (i.e., only highly reliable data can fully benefit from increasing sample sizes) has been demonstrated elsewhere in psychology (e.g., Jacobucci et al., 2020; McNamara et al., 2022). Though these papers need not be discussed extensively in the manuscript, they should at least be cited as this is a problem not specific to neuroimaging.

We would like to thank the reviewer for pointing out these references. We have integrated these into our discussion. We believe that our approach builds on these by providing an even more approachable examination of the issue by using data and datasets that

researchers are familiar with, rather than simulated relationships between a number of theoretical variables.

References

Bollen, K. A. (1989). The consequences of measurement error. *Structural equations with latent variables*, 151-178.

Spearman, C. (1910). Correlation calculated from faulty data. *British Journal of Psychology*, 3(3), 271–295.

Jacobucci, R., & Grimm, K. J. (2020). Machine learning and psychological research: The unexplored effect of measurement. *Perspectives on Psychological Science*, 15(3), 809-816.

McNamara, M. E., Zisser, M., Beevers, C. G., & Shumake, J. (2022). Not just “big” data: Importance of sample size, measurement error, and uninformative predictors for developing prognostic models for digital interventions. *Behaviour Research and Therapy*, 104086.

Reviewers' Comments:

Reviewer #1:

Remarks to the Author:

I appreciate the care with which the authors have addressed the concerns raised in the previous round of reviews. Overall I felt that they have largely addressed the concerns raised by myself and the other reviewers. I do have one remaining confusion, which I hope the authors can clarify.

In particular, it is unclear how Supplementary Figures 5 and 6 were generated. The text suggests that these were based on real data, but that can't be the case since the measures have reliability as high as 1. We know that all else being equal, the shorter scans should have lower reliability than the longer scans, so something was done in order to make them have a specific amount of reliability, but then it's unclear what differs between the short versus long scans.

Reviewer #3:

Remarks to the Author:

The authors did an excellent job addressing my concerns, very well done. The tone of the paper is much more balanced and includes a lot of important nuance. I think the neuroimaging community will enjoy this.

Reviewer #1 (Remarks to the Author):

I appreciate the care with which the authors have addressed the concerns raised in the previous round of reviews. Overall I felt that they have largely addressed the concerns raised by myself and the other reviewers. I do have one remaining confusion, which I hope the authors can clarify.

In particular, it is unclear how Supplementary Figures 5 and 6 were generated. The text suggests that these were based on real data, but that can't be the case since the measures have reliability as high as 1. We know that all else being equal, the shorter scans should have lower reliability than the longer scans, so something was done in order to make them have a specific amount of reliability, but then it's unclear what differs between the short versus long scans.

We thank the reviewer for their helpful feedback round that improved our manuscript considerably. We apologise for the confusion that our supplementary figures 5 and 6 caused. For consistency across figures in the manuscript, the x-axis in both plots represents the reliability of behaviour, not feature reliability or the joint reliability of both predicted behaviour and features. The reason why the plot in Supplementary Figures 5 and 6 ends at perfect reliability (i.e. 1.0) is because it displays the results of age (and simulated age) prediction presented in parts of the manuscript (e.g. Figure 1A) as a toy example with a variable that ought to have close to perfect reliability (also discussed in the main text). In contrast, supplementary figure 9 (pasted below for reference) shows the impact of feature reliability on the prediction accuracy of total cognition, here the plot does not end at 1.0 anymore, as it is corrected for the reliability of total cognition reported in previous literature. Feature reliability in Supplementary Figures 5, 6 and 9 was manipulated by reducing the amount of resting state fMRI data used to calculate functional connectivity (as this negatively impacts reliability - e.g.: Noble et al., 2017; Cho et al., 2021, Fig. 8) before being used for predictions (see also Methods section). To clarify this, we have greatly updated the legends for Supplementary Figures 5 and 6 (and 9):

“Supplementary Figure 5. Influence of feature reliability (functional connectivity) on the prediction of age with decreasing reliability. The x-axis represents the reliability of age, starting with empirically measured age at ICC \approx 1.0, followed by age with reduced reliability through simulations as in the main text. Feature reliability (represented by colours) was manipulated by reducing the amount of rsfMRI time course used to calculate functional connectivity, from the average of all four sessions collected on both days used in all main analyses to an average of 13 minutes collected on the first day only and finally, connectivity calculated from a single 6 minutes session collected on the first day in the anterior-to-posterior direction (same as only session in the UKB). Solid lines represent the mean prediction accuracy across all 100 simulated datasets in each reliability band, shaded areas represent 2 standard deviations in prediction accuracies.”

Supplementary Figure 6. Influence of feature reliability (functional connectivity) on the prediction of age with decreasing reliability. The x-axis represents the reliability of

age, starting with empirically measured age at ICC ≈ 1.0 , followed by age with reduced reliability through simulations as in the main text. Age prediction from functional connectivity was calculated on day 1 and 2 rsfMRI acquisition separately. Both days had 13 minutes and 22 seconds of data. Solid lines represent the mean prediction accuracy across all 100 simulated datasets in each reliability band, shaded areas represent 2 standard deviations in prediction accuracies.

Supplementary Figure 9:

Supplementary Figure 9. Influence of feature reliability (functional connectivity) on prediction of total cognition with decreasing reliability. The x-axis represents the reliability of the NIH Toolbox total cognition composite score, starting at ICC = 0.9 based on Heaton et al. (2014), followed by total cognition with reduced reliability through simulations as in the main text. Feature reliability (represented by colours) was manipulated by reducing the amount of rsfMRI time course used to calculate functional connectivity, from the average of all four sessions collected on both days used in all main analyses to an average of 13 minutes collected on the first day only and finally, connectivity calculated from a single 6-minute session collected on the first day in the anterior-to-posterior direction (same as only session in the UKB). Solid lines represent the mean prediction accuracy across all 100 simulated datasets in each reliability band, shaded areas represent 2 standard deviations in prediction accuracies.

Reviewer #3 (Remarks to the Author):

The authors did an excellent job addressing my concerns, very well done. The tone of the paper is much more balanced and includes a lot of important nuance. I think the neuroimaging community will enjoy this.

We thank the reviewer for their helpful feedback, which considerably improved our manuscript!

References

Cho, J. W., Korchmaros, A., Vogelstein, J. T., Milham, M. P., & Xu, T. (2021). Impact of concatenating fMRI data on reliability for functional connectomics. *NeuroImage*, 226, 117549. <https://doi.org/10.1016/j.neuroimage.2020.117549>

Noble, S., Spann, M. N., Tokoglu, F., Shen, X., Constable, R. T., & Scheinost, D. (2017). Influences on the Test-Retest Reliability of Functional Connectivity MRI and its Relationship with Behavioral Utility. *Cerebral Cortex*, 27(11), 5415-8211. <https://doi.org/10.1093/cercor/bhx230>